# Decomposing and Interpreting Image Representations via Text in ViTs Beyond CLIP

**Sriram Balasubramanian**
Department of Computer Science
University of Maryland, College Park
sriramb@cs.umd.edu

**Samyadeep Basu**
Department of Computer Science
University of Maryland, College Park
sbasu12@umd.edu

**Soheil Feizi**
Department of Computer Science
University of Maryland, College Park
sfeizi@cs.umd.edu

## Abstract

Recent work has explored how individual components of the CLIP-ViT model contribute to the final representation by leveraging the shared image-text representation space of CLIP. These components, such as attention heads and MLPs, have been shown to capture distinct image features like shape, color or texture. However, understanding the role of these components in arbitrary vision transformers (ViTs) is challenging. To this end, we introduce a general framework which can identify the roles of various components in ViTs beyond CLIP. Specifically, we (a) automate the decomposition of the final representation into contributions from different model components, and (b) linearly map these contributions to CLIP space to interpret them via text. Additionally, we introduce a novel scoring function to rank components by their importance with respect to specific features. Applying our framework to various ViT variants (e.g. DeiT, DINO, DINOv2, Swin, MaxViT), we gain insights into the roles of different components concerning particular image features. These insights facilitate applications such as image retrieval using text descriptions or reference images, visualizing token importance heatmaps, and mitigating spurious correlations. We release our code to reproduce the experiments at https://github.com/SriramB-98/vit-decompose

## 1 Introduction

Vision transformers and their variants [10, 22, 7, 33, 17, 32] have emerged as powerful image encoders, becoming the preferred architecture for modern image foundation models. However, the mechanisms by which these models transform images into representation vectors remain poorly understood. Recently, Gandelsman et al. [11] made significant progress on this question for CLIP-ViT models with two key insights: (i) They demonstrated that the residual connections and attention mechanisms of CLIP-ViT enable the model output to be mathematically represented as a sum of vectors over layers, attention heads, and tokens, along with contributions from MLPs and the CLS token. Each vector corresponds to the contribution of a specific token attended to by a particular attention head in a specific layer. (ii) These contribution vectors exist within the same shared image-text representation space, allowing the CLIP text encoder to interpret each vector individually via text.

Extending this approach to other transformer-based image encoders presents several challenges. Popular models like DeiT [32], DINO-ViT [7, 22], and Swin [17] lack a corresponding text encoder to

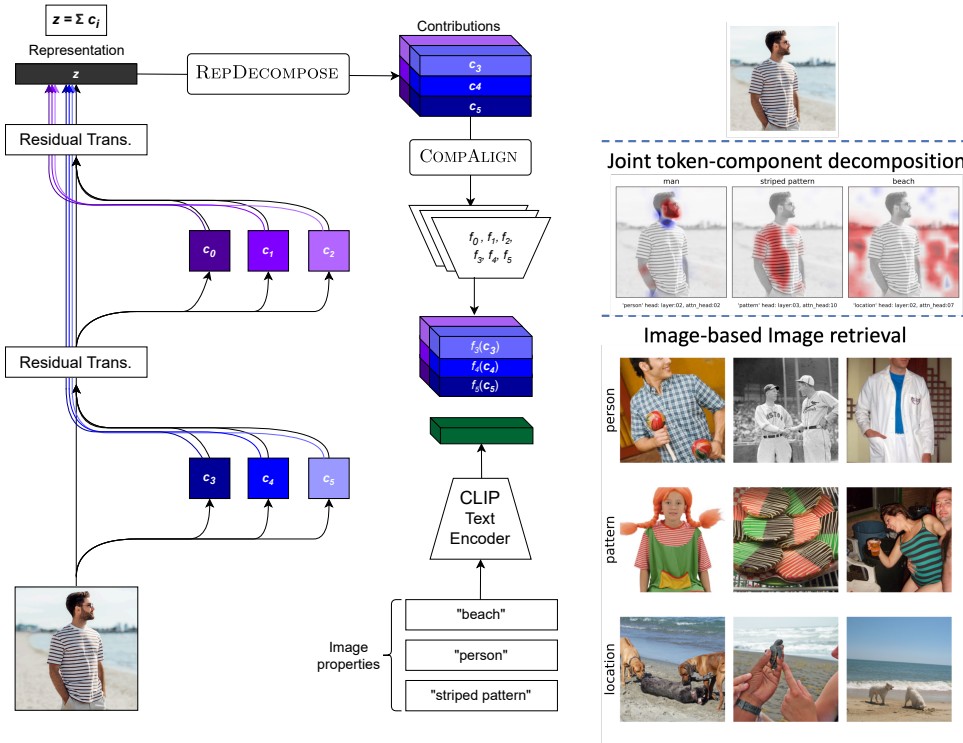

Figure 1: **(Left) Workflow:** The first step (REPDECOMPOSE) is to decompose a representation $z$ into contributions from its model components $c_i$ after being transformed by residual transformations like LayerNorm, linear projections, resampling, patch merging and so on. The second step (COMPALIGN) aligns each contribution to CLIP space using a set of linear maps $f_0, f_1, \ldots, f_n$ on the corresponding contributions $c_0, c_1, \ldots, c_n$. We can then interpret these aligned contributions using the CLIP text encoder. **(Right) Applications of our method:** (a) Visualizing contributions of each token through a specific component using a joint token-component decomposition (b) Retrieving images that are close matches of the reference image (on top) with respect to a given image feature like pattern, person, or location

interpret the component contributions. Additionally, extracting the contribution vectors corresponding to these components is not straightforward, as they are often not explicitly computed during the forward pass of the model. Other complications include diverse attention mechanisms such as grid attention, block attention (in MaxViT), and windowed/shifted windowed attention (in Swin), as well as various linear transformations like pooling, downsampling, and patch merging applied to the residual streams between attention blocks. These differences necessitate a fresh mathematical analysis for each model architecture, followed by careful application of necessary transformations to the intermediate output of each component to determine its contribution to the final representation. To address these challenges, we propose our framework (described in Figure 1) to identify roles of components in general ViTs.

**First**, we *automate the decomposition of the representation* by leveraging the computational graph created during the forward pass. This results in a drop-in function, REPDECOMPOSE, that can decompose any representation into contributions vectors from model components simply by calling it on the representation. Since this method operates on the computational graph, it is agnostic to the specifics of the model implementation and thus applicable to a variety of model architectures.

**Secondly**, we introduce an algorithm, COMPALIGN, to *map each component contribution vector to the image representation space of a CLIP model*. We train these linear maps with regularizations so that these maps preserve the roles of the individual components while also aligning the model's image representation with CLIP's image representation. This allows us to map each contribution

vector from any component to CLIP space, where they can be interpreted through text using a CLIP text encoder.

**Thirdly**, we observe that there is often *no straightforward one-to-one mapping* between model components and common image features such as shape, pattern, color, and texture. Sometimes, a single component may encode multiple features, while multiple components may be required to fully encode a single feature. To address this, we propose a *scoring function* that assigns an importance score to each component-feature pair. This allows us to rank components based on their importance for a given feature, and rank features based on their importance for a given component.

Using this ranking, we proceed to analyze diverse vision transformers such as DeiT, DINO, Swin, and MaxViT, in addition to CLIP, in terms of their components and the image features that they are responsible for encoding. We consistently find that many components in these models encode the same feature, particularly in ImageNet pre-trained models. Additionally, individual components in larger models MaxVit and Swin do not respond to any image feature strongly, but can encode them effectively in combination with other components. This diffuse and flexible nature of feature representation underscores the need for interpreting them using a continuous scoring and ranking method as opposed to labelling each component with a well-defined role. We are thus able to perform tasks such as image retrieval, visualizing token contributions, and spurious correlation mitigation by carefully selecting or ablating specific components based on their scores for a given property.

## 2   Related Work

Several studies attempt to elucidate model predictions by analyzing either a subset of input example through heatmaps [27, 30, 31, 18] or a subset of training examples [15, 23, 24]. Nevertheless, empirical evidence suggests that these approaches are often unreliable in real-world scenarios [14, 3]. These methods do not interpret model predictions in relation to the model's internal mechanisms, which is essential for gaining a deeper understanding of the reliability of model outputs.

**Internal Mechanisms of Vision Models:** Our work is closely related to the studies by Gandelsman et al. [11] and Vilas et al. [34], both of which analyze vanilla ViTs in terms of their components and interpret them using either CLIP text encoders or pretrained ImageNet heads. Like these studies, our research can be situated within the emerging field of representation engineering [36] and mechanistic interpretability [6, 5]. Other works [4, 12, 21] focus on interpreting individual neurons to understand vision models' internal mechanisms. However, these methods often fail to break down the model's output into its subcomponents, which is crucial for understanding model reliability. Shah et al. [29] examine the direct effect of model weights on output, but do not study the fine-grained role of these components in building the final image representation. Balasubramanian and Feizi [1] focus on expressing CNN representations as a sum of contributions from input regions via masking.

**Interpreting models using CLIP:** Many recent works utilize CLIP [25] to interpret models via text. Moayeri et al. [19] align model representations to CLIP space with a linear layer, but it is limited to only the final representation and can not be applied to model components. Oikarinen and Weng [20] annotate individual neurons in CNNs via CLIP, but their method cannot be extended easily to high-dimensional component vectors. COMPALIGN is related to model stitching in which one representation space is interpreted in terms of another by training a map between two spaces [2, 16].

## 3   Decomposing the Final Image Representation

Recently, Gandelsman et al. [11] decomposed $z_{\text{CLS, fin}}$, the final `[CLS]` representation of the CLIP's image encoder as a sum over the contributions from its attention heads, layers and token positions, as well as contributions from the MLPs. In particular, they observe that the last few attention layers have a significant direct impact on the final representation. Thus, this representation can be decomposed as: $z_{\text{CLS, fin}} = z_{\text{CLS, init}} + \sum_{l=1}^{L} c_{l,\text{MLP}} + \sum_{l=1}^{L} \sum_{h=1}^{H} \sum_{t=1}^{N} c_{l,h,t}$, where $L$, $H$, $N$ correspond to the number of layers, number of attention heads and number of tokens. Here, $c_{l,h,t}$ denotes the contribution of token $t$ though attention head $h$ in layer $l$, while $c_{l,\text{MLP}}$ denotes the contribution from the MLP in layer $l$. Due to this linear decomposition, different dimensions can be reduced by summing over them to identify the contributions of tokens or attention heads to the final representation. While this decomposition is relatively simple for vanilla ViTs, it cannot be directly used for general ViT architectures due to use of self-attention variants such as window attention, grid attention, or block

---

**Algorithm 1** REPDECOMPOSE

---

**Input:** $\boldsymbol{f}$, the final node (denoting the function that output the representation $\boldsymbol{z}$) in the computational graph $G$

**Output:** $\{\boldsymbol{c}^j\}_{j=1}^{N}$, $N$ direct contributions from various components (indexed by $j$) in the model after graph traversal

**function** REPDECOMPOSE($\boldsymbol{f}$)
    $\boldsymbol{z} = \boldsymbol{f}(\boldsymbol{z}_1, \boldsymbol{z}_2, \ldots, \boldsymbol{z}_n)$
    **if** $\boldsymbol{f}$ is non-linear **then**
        **return** $[\boldsymbol{z}]$                           ▷ *Cannot decompose further*
    **else**                                        ▷ $\boldsymbol{f}$ *is linear*
        Let $\boldsymbol{z}_1 = \boldsymbol{f}_1(\ldots), \boldsymbol{z}_2 = \boldsymbol{f}_2(\ldots), \ldots, \boldsymbol{z}_n = \boldsymbol{f}_n(\ldots)$
        $\forall i, \{\boldsymbol{c}_i^j\}_{j=1}^{N_i} = $ REPDECOMPOSE($\boldsymbol{f}_i$) ▷ $\boldsymbol{z}_i = \sum_{j=1}^{N_i} \boldsymbol{c}_i^j$ ( $\boldsymbol{c}_i^j$ *are component contributions* )
        Then, $\boldsymbol{z} = \boldsymbol{f}(\sum_{j=1}^{N_1} \boldsymbol{c}_1^j, \sum_{j=1}^{N_2} \boldsymbol{c}_2^j, \ldots, \sum_{j=1}^{N_n} \boldsymbol{c}_n^j)$
        Or, $\boldsymbol{z} = \sum_{i=1}^{n} \sum_{j=1}^{N_i} \boldsymbol{f}'(\boldsymbol{c}_i^j)$      ▷ $f'$ *exists since* $\boldsymbol{f}$ *distributes over inputs due to linearity*
        **return** $[\{f'(\boldsymbol{c}_i^j)\}_{j=1}^{N_i}]_{i=1}^{n}$

---

attention, combined with operations such as pooling or patch merging on the residual stream. The final representation may also not just be a single $\boldsymbol{z}_{\text{CLS, fin}}$ but $\frac{1}{N}\sum_{i=1}^{N} \boldsymbol{z}_{i,\text{fin}}$ or even $\frac{1}{L}\sum_{i=1}^{L} \boldsymbol{z}_{\text{CLS},i}$, or some combination of the above.

### 3.1 REPDECOMPOSE: Automated Representation Decomposition for ViTs

We thus seek a general algorithm which can automatically decompose the representation for general ViTs. This can be done via a recursive traversal of the computation graph. Suppose the final representation $\boldsymbol{z}$ can be decomposed into component contributions $\boldsymbol{c}_{i,t}$ such that $\boldsymbol{z} = \sum_{i,t} \boldsymbol{c}_{i,t}$. Here each $\boldsymbol{c}_{i,t}$ corresponds to the contribution of a particular token $t$ through some model component $i$. For convenience, let $\boldsymbol{c}_i = \sum_t \boldsymbol{c}_{i,t}$ . Then, if given access to the computational graph of $\boldsymbol{z}$, we can identify potential linear components $\boldsymbol{c}_{i,t}$ by recursively traversing the graph starting from the node which outputs $\boldsymbol{z}$ in reverse order till we hit a non-linear node. The key insight here is that the output of any node which performs a linear reduction (defined as a linear operation which results in a reduction in the number of dimensions) is equivalent to a sum of individual tensors of the same dimension as the output. These tensors can be collected and transformed appropriately during the graph traversal to obtain a list of tensors $\boldsymbol{c}_{i,t}$, each members of the same vector space as the representation $\boldsymbol{z}$. This kind of linear decomposition is possible due to the overwhelmingly linear nature of transformers. The high-level logic of REPDECOMPOSE is detailed in Algorithm 1, please refer to Algorithm 2 in the appendix or the code for a more detailed description. We also illustrate the operation of the algorithm on an attention-MLP block in the appendix.

In practice, the number of components quickly explodes as there are a very large number of potential component divisions for a given model. To make analysis and computation tractable, we restrict it to only the attention heads and MLPs with no finer divisions. We also constrain REPDECOMPOSE to only return the *direct* contributions of these components to the output. This means that the contribution $\boldsymbol{c}_i$ is the *direct* contribution of component $i$ to $\boldsymbol{z}$, and does not include its contribution to $\boldsymbol{z}$ via a downstream component $j$. Additionally, the token $t$ in $\boldsymbol{c}_{i,t}$ is present in the input of the component $i$, and not the input image. In principle, REPDECOMPOSE could return higher order terms such as $\boldsymbol{c}_{j,i}$ which is the contribution of model component $i$ via the downstream component $j$. A full understanding of these higher order terms is essential to get a complete picture of the inner mechanism of a model, however we defer this for future work.

## 4 Aligning the component representations to CLIP space

Having decomposed the representation into contributions from relevant model components, we now aim to interpret these contributions through text using CLIP by mapping them to CLIP space. Formally, given that we have a set of vectors $\{\boldsymbol{c}_i\}_{i=1}^{N}$ such that $\sum_i^N \boldsymbol{c}_i = \boldsymbol{z}$, the final representation of model, we require a set of linear maps $f_i$ such that the sum of $\sum_i f_i(\boldsymbol{c}_i) = \boldsymbol{z}_{\text{CLIP}}$, the final

| Embedding Source | ImageNet pretrained | One map only | COMPALIGN ($\lambda = 0$) | COMPALIGN |
|---|---|---|---|---|
| TEXTSPAN's top 10 descriptions of a random component | wardrobe
medicine cabinet
window shade
desk
barbershop
refrigerator
library
shoji screen
bathtub
dining table | gyromitra
home theater
drumstick
Samoyed
muzzle
bookstore
dining table
medicine cabinet
park bench
tusker | bookcase
snorkel
red wolf
barbershop
microwave oven
bassinet
disc brake
dining table
sink
window screen | filing cabinet
snorkel
bakery
bathtub
dining table
red wolf
gyromitra
shoji screen
Norwich Terrier
bookstore |
| Match rate | - | 0.08 | 0.155 | 0.185 |
| Cosine Distance | - | 0.23 | 0.18 | 0.17 |

Table 1: Comparison of different methods to map the representation space of ImageNet-1k pre-trained DeiT-B/16 to CLIP image representation space. The green colored texts are exact matches with the top-10 descriptions obtained from the imagenet pretrained embeddings, while the orange colored texts are approximate matches. The match rate is the average fraction of exact matches across all components, while cosine distance is the average cosine distance between the CLIP representations and the transformed model representations on ImageNet

representation of the CLIP model. Once we have these CLIP aligned vectors, we can proceed to interpret them via text using CLIP's text encoders.

However, from an interpretability standpoint, a few additional constraints on the linear maps are desirable. Consider a component contribution $c_i$ and two directions $u, v$ belonging to the same vector space as $c_i$ which represent two distinct features, say shape and texture. Let us further assume that the component's dominant role is to identify shape, and thus the variance of the projection of $c_i$ along $u$ is higher than that of $v$. We want this relative importance of features to be maintained in $f_i(c_i)$. Additionally, we also want any two linear maps $f_i$ and $f_j$ to not change the relative norms of features in components $c_i$ and $c_j$. We can express these conditions formally as follows:

1. **Intra-component norm rank ordering**: For any two vectors $u, v$ and a linear map $f_i$ such that $\|u\| \leq \|v\|$, we have $\|f_i(u)\| \leq \|f_i(v)\|$

2. **Inter-component norm rank ordering**: For any two vectors $u, v$ and linear maps $f_i, f_j$ such that $\|u\| \leq \|v\|$, we have $\|f_i(u)\| \leq \|f_j(v)\|$

**Theorem 1.** *Both of the above conditions together imply that all linear maps $f_i$ must be a scalar multiple of an orthogonal transformation, that is for all $i$, $f_i^T f_i = kI$ for some constant $k$. Here, $I$ is the identity transformation.*

The proof is deferred to appendix E. We can now formulate a novel alignment method, COMPALIGN, to map contributions of model components to CLIP space. COMPALIGN minimizes a loss function over $\{f_i\}_{i=1}^N$ to obtain a good alignment between model representation space and CLIP space:

$$L(\{f_i\}_{i=1}^N) = \mathbb{E}_{\{c_i\}_{i=1}^N, z_{\text{CLIP}}} \left[ 1 - \cos\left( \sum_i f_i(c_i), z_{\text{CLIP}} \right) \right] + \lambda \sum_i \|f_i^T f_i - I\|_F$$

The first term of the objective is the *alignment loss*, which is the average cosine distance between the CLIP representation $z_{\text{CLIP}}$ and the transformed model representation $\sum_i f_i(c_i)$. It quantifies the directional discrepancy between the two vectors. The second term is the *orthogonality regularizer* which imposes a penalty if the linear maps $f_i$ are not orthogonal, ensuring that the $f_i$ adhere closely to the specified conditions. We can now train $f_i$ using the above loss function on ImageNet-1k. The training is label-free and can be done even over unlabeled datasets. We obtain $z_{\text{CLIP}}$ from the CLIP image encoder and $\{c_i\}_{i=1}^N$ from running REPDECOMPOSE on the final representation of the model.

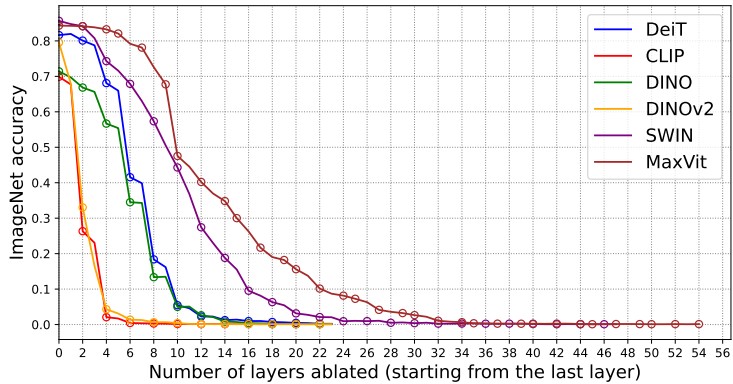

Figure 2: Ablation results for various different image encoders. The top-1 ImageNet accuracy is plotted as the layers of the model are increasingly ablated away, starting from the last layer up till the first layer. The circles on the plot represent the endpoints of blocks, the definition of which varies across model architectures. For the vanilla ViT variants, a block is an attention MLP pair, while for SWIN, it is a pair of windowed/shifted windowed attention and an MLP. For MaxVit, this might either be a grid/block attention-MLP pair, or an MBConv block.

**Ablation study:** We now conduct an ablation study on COMPALIGN. The first naive alignment method is the case where all $f_i$ are the same linear map $f$, without constraints on $f$, similar to [19]. The second method is a version of COMPALIGN with $\lambda = 0$, where all $f_i$ are different but not trained with the orthogonality regularizer. To compare these methods, we first get a "ground truth" description for each model component by using the TEXTSPAN [11] algorithm on the class embedding vectors from the ImageNet pre-trained head. TEXTSPAN retrieves those class embedding vectors along which variance of the component output is maximized, thus yielding descriptions of each component in terms of the top 10 most dominant ImageNet classes. We then use COMPALIGN and the two baselines to map the representations to CLIP space, and apply TEXTSPAN on CLIP embedded ImageNet class vectors to label each model component. We can then compare the descriptions this yields with the "ground truth" text description for each head. The results, shown in Tab. 1, indicate that COMPALIGN's TEXTSPAN descriptions have the most matches to the ImageNet pre-trained descriptions, followed by COMPALIGN with $\lambda = 0$ and the naive single map method. This trend is similar in the average cosine distance between the CLIP representations and the transformed model representations.

## 5   Component ablation

To identify the most relevant model layers for downstream tasks, we progressively ablate them and measure the drop in ImageNet classification accuracy. Ablation involves setting a layer's contribution to its mean value over the dataset. We use the following models from Huggingface's `timm` [35] repository: (i) DeiT (ViT-B-16) [32], (ii) DINO (ViT-B-16) [7], (iii) DINOv2 (ViT-B-14) [22], (iv) Swin Base (patch size = 4, window size = 7) [17], (v) MaxViT Small [33], along with (vi) CLIP (ViT-B-16) [9] from `open_clip` [13]. DeiT, Swin, and MaxViT are pretrained on ImageNet with a supervised classification loss, DINO on ImageNet with a self-supervised loss, DINOv2 on LVD-142M with a self-supervised loss, while CLIP is pretrained on a LAION-2B subset with contrastive loss.

In Fig. 2, we see that for models not trained on ImageNet (CLIP and DINOv2), removing the last few layers quickly drops the accuracy to zero. In contrast, models trained on ImageNet experience a more gradual decline in accuracy, reaching zero only after more than half the layers are ablated. This trend is consistent across both self-supervised (DINO) and classification-supervised (DeiT, SWIN, MaxViT) models. This suggests that ImageNet-trained models encode useful features redundantly across layers for the classification task. Additionally, larger models with more layers, such as MaxVit, show significantly more redundancy, with minimal accuracy impact from ablating the last four layers. Conversely, the first few layers in all models contribute little to the output. Therefore, our analysis in the subsequent sections is focused on the last few layers of each model.

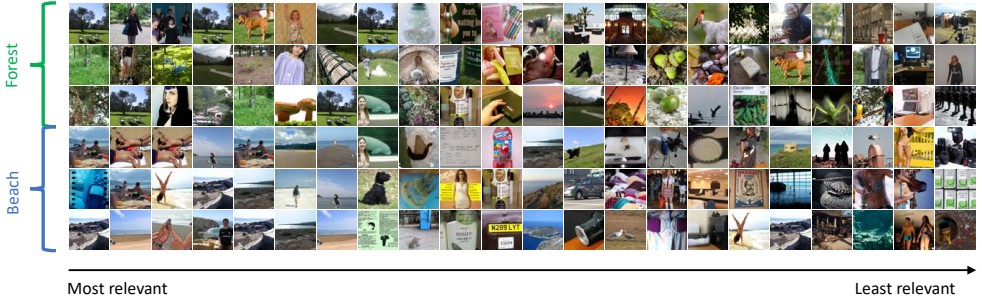

Most relevant                                                                Least relevant

Figure 3: Top-3 images retrieved by DeiT components for "forest" and "beach" ordered according to their relevance for the attribute "location". Each column here corresponds to the images returned by the sum of contributions of 3 components, so column $i$ corresponds to components $c_{3i}, c_{3i+1}, c_{3i+2}$. A large fraction of components which can recognize the "location" feature are sorted correctly by the scoring function

## 6 Feature-based component analysis

We now analyze the final representation in terms finer components like attention heads and MLPs, focusing on the last few significant layers. We limit decomposition to 10 layers for DeiT, DINO, and DINOv2, but 12 layers for SWIN and 20 layers for MaxVit due to their greater depth and redundancy across components. We accumulate contributions from the remaining components in a single vector $c_{\text{init}}$, expressing $z$ as $c_{\text{init}} + \sum_i^N c_i$, where $N+1$ is the total number of components including $c_{\text{init}}$. Here, $N = 65$ for DeiT, DINO, and DINOv2; $N = 134$ for SWIN, and $N = 156$ for MaxVit.

We then ask if it is possible to attribute a feature-specific role to each component using an algorithm such as TEXTSPAN [11]. These image features may be low-level (shape, color, pattern) or high-level (such as location, person, animal). However, such roles are not necessarily localized to a single component, but may be distributed among multiple components. Furthermore, each individual component by itself may not respond significantly to a particular feature, but it may jointly contribute to identifying a feature along with other components. Thus, rather than rigidly matching each component with a role, we aim to devise a *scoring function* which can assign a score to each component - feature pair, which signifies of how important the component is for identifying a given image feature. A continuous scoring function allows us to select multiple components relevant to the feature by sorting the components according to their score.

We devise this scoring function (described in the appendix in Alg. 3) by looking at the projection of each contribution vector $c_i$ onto a vector space corresponding to a certain feature. Suppose we have a feature, "pattern", that we want to attribute to the components. We first describe the feature in terms of an example set of feature *instantiations*, such as "spotted", "striped", "checkered", and so on. We then embed each of these texts to CLIP space, obtaining a set of embeddings $B$. We also calculate the CLIP aligned contributions $f_i(c_i)$ for each component $i$ over an image dataset (ImageNet-1k validation split). Then, the score is simply the correlation between projection of $f_i(c_i)$ and the projection of $\sum_i f_i(c_i)$ onto the vector space spanned by $B$. Intuitively, this measures how closely the component's contribution correlates with the overall representation. The

| Model | Feature ordering | Component ordering |
|---|---|---|
| DeiT | 0.531 | 0.684 |
| DINO | 0.714 | 0.723 |
| DINOv2 | 0.716 | 0.703 |
| SWIN | 0.628 | 0.801 |
| MaxVit | 0.681 | 0.849 |

Table 2: Spearman's rank correlation between the orderings induced by CLIP score and component score averaged over a selection of common features

scores obtained for each component and feature can be used to rank the components according to its importance for a given feature to obtain a *component ordering*, or to rank the features according to its importance for a specific component to get a *feature ordering* .

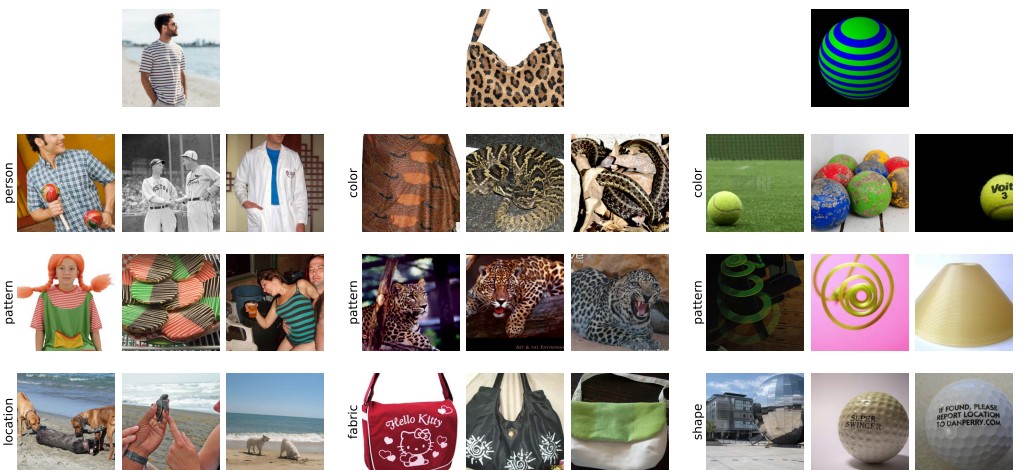

Figure 4: Top-3 images retrieved by the most significant components for various features relevant to the reference image (displayed on top). The models used are (from left to right) DINO, DeiT, and SWIN. More exhaustive results can be found in appendix H

## 6.1 Text based image retrieval

We can now use our framework to identify components which can retrieve images possessing a certain feature most effectively. Using the scoring function described above, we can identify the top $k$ components $\{c_i\}_{i=1}^k$ which are the most responsive to a given feature $p$. We can use the cosine similarity of $\sum_{i=1}^k f_i(c_i)$ to the CLIP embedding of an instantiation $s_p$ of the feature $p$ to retrieve the closest matches in ImageNet-1k validation split. In Fig. 3, we show the top 3 images retrieved by different components of the DeiT model for the location instantiation "forest" and "beach" when sorted according to the component ordering for the "location" feature. As the component score decreases, the images retrieved by the components grow less relevant. Also note that a significant fraction of components are capable of retrieving relevant images. This further confirms the need for a continuous scoring function which can identify multiple components relevant to a feature.

To quantitatively verify our scoring function, we devise the following experiment. We first choose a set of common image features such as color, pattern, shape, and location, with a representative set of feature instantiations for each (details in appendix B). The scoring function induces a component ordering for each feature $p$ and feature ordering for each component $i$. We then compute the cosine similarity $\text{sim}_{i,s_p} = \cos(f_i(c_i), \boldsymbol{y}_{s_p,\text{CLIP}})$ where $\boldsymbol{y}_{s_p,\text{CLIP}}$ is the CLIP text embedding of $s_p$. We can compare this to the cosine similarity $\text{sim}_{\text{CLIP},s_p} = \cos(\boldsymbol{z}_{\text{CLIP}}, \boldsymbol{y}_{s_p,\text{CLIP}})$ where $\boldsymbol{z}_{\text{CLIP}}$ is the CLIP image representation. The correlation coefficient between $\text{sim}_{i,s_p}$ and $\text{sim}_{\text{CLIP},s_p}$ over an image dataset can be viewed as another score which is purely a function of how well the component $i$ can retrieve images matching $s_p$ as judged by CLIP. Averaging this correlation coefficient over all $s_p$ for a given $p$ yields a "ground truth" proxy for our scoring function. We can measure the Spearman rank correlation (which ranges from -1 to 1) between the component (or feature) ordering induced by our scoring function and the ground truth and average it over features (or components). In Tab. 2, we observe that the rank correlation is significantly high for all models for both feature and component ordering. The individual rank correlations for component orderings for common features can be found in Tab. 4.

## 6.2 Image based image retrieval

We can also retrieve images that are similar to a reference image with respect to a specific feature. To do this, we first choose components which are highly significant for the given feature while being comparatively less relevant for other features. Mathematically, for a feature $p \in P$, the set of all relevant features, we want to choose component $i$ with score $s_{i,p}$ such that the quantity $\min_{p' \in P \backslash p} s_{i,p} - s_{i,p'}$ is maximised. Intuitively, we want components which have the highest

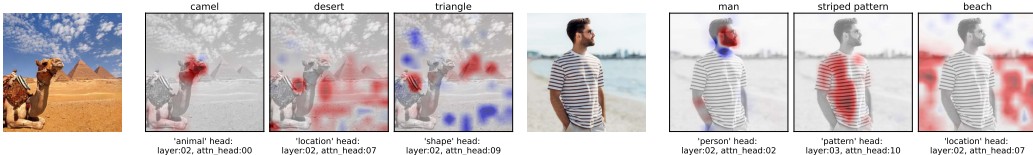

Figure 5: Visualization of token contributions as heatmaps for two example images for the DeiT model. The relevant feature and the head most closely associated with the feature is displayed on the bottom of the heatmap, while the feature instantiation is displayed on the top. The layer numbering starts from the last layer (which has index '00'). The regions highlighted in red contribute positively to the prediction, while blue regions contribute negatively. More results in appendix I

gap between $s_{i,p}$ and $s_{i,p'}$ where $p'$ can be any other feature. We can then select a set of $k$ such components $C_k$ by sorting over the score gap, and sum them to obtain a feature-specific image representation $z_p = \sum_{i \in C_k} c_i$ . Now, we can retrieve any image $x'$ similar in feature $p$ to a reference image $x$ by computing the cosine similarity between $z'_p$ and $z_p$, which are the feature-specific image representations for $x'$ and $x$. We show a few examples for image based image retrieval in Fig. 4. Here, we tune $k$ to ensure that it is not so small that the retrieved images do not resemble the reference image at all, and not so large that the retrieved images are overall very similar to the reference image. We can see that the retrieved images are significantly similar to the reference image with respect to the given feature, but not similar overall. For example, when the reference image is a handbag with a leopard print, the "pattern" components retrieve images of leopards which have the same pattern, while the "fabric" components return other bags which are made of similar glossy fabric. Similarly, for the ball with a spiral pattern on it, we retrieve images which resemble the spiral pattern in the second row, while they resemble the shape in the third row.

Note that this experiment only involves the alignment procedure for computing the scores and thereby selecting the component set $C_k$. The process of retrieving the images is based on $z_p$ which exists in the model representation space and not CLIP space. This shows that the model inherently has components which (while not constrained to a single role) are specialized for certain properties, and this specialization is not a result of the CLIP alignment procedure.

## 6.3 Visualizing token contributions

As discussed in Section 3.1, contribution from a component $i$ can be further decomposed as a sum over contributions from a tokens, so $c_i = \sum_t c_{i,t}$. For any particular CLIP text embedding vector $u$ corresponding to a realization of some feature $p$, we have $u^\top f_i(c_i) = \sum_t u^\top f_i(c_{i,t})$. We can visualize this token-wise score $u^\top f_i(c_{i,t})$ as a heat map to know which tokens are the most influential with respect to $u$. We show the heat map obtained via this procedure in Fig. 5 for two example images for the DeiT model. The components used for each heat map correspond to the feature being highlighted and are selected using the scoring function we described previously. We can observe that the heatmaps are localized within image portions which correspond to the text description. We also compare our method

| Model name | Worst group accuracy | Average group accuracy |
|---|---|---|
| DeiT | $0.733 \rightarrow \mathbf{0.815}$ | $0.874 \rightarrow \mathbf{0.913}$ |
| CLIP | $0.507 \rightarrow \mathbf{0.744}$ | $0.727 \rightarrow \mathbf{0.790}$ |
| DINO | $0.800 \rightarrow \mathbf{0.911}$ | $0.900 \rightarrow \mathbf{0.938}$ |
| DINOv2 | $0.967 \rightarrow \mathbf{0.978}$ | $0.983 \rightarrow \mathbf{0.986}$ |
| SWIN | $0.834 \rightarrow \mathbf{0.871}$ | $0.927 \rightarrow \mathbf{0.944}$ |
| MaxVit | $0.777 \rightarrow \mathbf{0.814}$ | $0.875 \rightarrow \mathbf{0.887}$ |

Table 3: Worst group accuracy and average group accuracy for Waterbirds dataset before and after intervention for various models (format is before $\rightarrow$ **after**)

against zero-shot segmentation methods for ImageNet classes such as GradCAM [28] and Chefer et al. [8] and find that our method outperforms the baselines (see Appendix J).

## 6.4 Zero-shot spurious correlation mitigation

We can also use the scoring function to mitigate spurious correlations in the Waterbirds dataset [26] in a zero-shot manner. Waterbirds dataset is a synthesized dataset where images of birds commonly

found in water ("waterbirds") and land ("landbirds") are cut out and pasted on top of images of land and water background. For this experiment, we regenerate the Waterbirds dataset following Sagawa et al. [26] but take care to discard background images with birds and thus eliminate label noise. We select the top 10 components for each model which are associated with the "location" feature but not with the "bird" class following the method we used in Sec. 6.2. We then ablate these components by setting their value to their mean over the Waterbirds dataset. In Tab. 3, we observe a significant increase in the worst group accuracy for all models, accompanied with an increase in the average group accuracy as well. The changes in all four groups can be found in appendix K in Tab. 6.

# 7 Conclusion

In this work, we propose a ViT component interpretation framework consisting of an automatic decomposition algorithm (REPDECOMPOSE) to break down the model's final representation into component contributions and a method (COMPALIGN) to map these contributions to CLIP space for text-based interpretation. We also introduce a continuous scoring function to rank components by their importance in encoding specific features and to rank features within a component. We demonstrate the framework's effectiveness in applications such as text-based and image-based retrieval, visualizing token-wise contribution heatmaps, and mitigating spurious correlations in a zero-shot manner.

## Acknowledgments

This project was supported in part by a grant from an NSF CAREER AWARD 1942230, ONR YIP award N00014-22-1-2271, ARO's Early Career Program Award 310902-00001, HR00112090132 (DARPA/ RED), HR001119S0026 (DARPA/ GARD), Army Grant No. W911NF2120076, the NSF award CCF2212458, NSF Award No. 2229885 (NSF Institute for Trustworthy AI in Law and Society, TRAILS), an Amazon Research Award and an award from Capital One.

## Author contributions

Sriram Balasubramanian conceived the main ideas, implemented the algorithms, conducted the experiments, and contributed to writing the paper. Samyadeep Basu contributed to the writing and provided essential advice on the presentation and direction of the paper. Soheil Feizi offered critical guidance on the presentation, writing, and overall direction of the paper.

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

# A  Limitations

Our analysis is limited in several ways which we hope to address in future work. Firstly, similar to Gandelsman et al. [11], we only consider the direct contributions from the last few layers, and do not look at the indirect contributions though other components. Secondly, we limit ourselves to decomposition only over attention heads and tokens, while convolutional blocks are not decomposed even if they might admit one. Furthermore, it is still unclear if we can identify certain directions or vector subspaces in the model component contributions which are strongly associated with a certain property. We believe that a detailed analysis of higher order contributions with a more fine-grained decomposition may be key for addressing these challenges.

# B  Implementation details

**Feature instantiations:**  We use the following features and corresponding feature instantiations. They are chosen arbitrarily:

1. **color**: "blue color", "green color", "red color", "yellow color", "black color", "white color"
2. **texture**: "rough texture", "smooth texture", "furry texture", "sleek texture", "slimy texture", "spiky texture", "glossy texture"
3. **animal**: "camel", "elephant", "giraffe", "cat", "dog", "zebra", "cheetah"
4. **person**: "face", "head", "man", "woman", "human", "arms", "legs"
5. **location**: "sea", "beach", "forest", "desert", "city", "sky", "marsh"
6. **pattern**: "spotted pattern", "striped pattern", "polka dot pattern", "plain pattern", "checkered pattern"
7. **shape**: "triangular shape", "rectangular shape", "circular shape", "octagon"
8. **fabric**: "linen", "velvet", "cotton", "silk", "chiffon"

**Hyperparameters:** The aligners are trained with learning rate $= 3 \times 10^{-4}$ , $\lambda = 1/768$ using the Adam optimizer (with default values for everything else) for upto an epoch on ImageNet validation split. Hyperparameters were loosely tuned for the DeiT model using the cosine similarity as a metric, and then fixed for the rest of the models. We may achieve better performance with more rigorous tuning. The number of components $k$ used for the image-based image retrieval experiment was tuned on an image-by-image basis. It is approximately around 9 for larger models like Swin or MaxVit, and around 3 for the rest.

**Computational resources:** The bulk of computation is utilized to compute component contributions and train the aligner. Most of the experiments in the paper were conducted on a single RTX A5000 GPU, with 32GB CPU memory and 4 compute nodes.

# C More detailed pseudocode for REPDECOMPOSE

---

**Algorithm 2** REPDECOMPOSE

---

**Input:** $z$, the final representation output by the model and the final node in the computational graph. $z.f$ is the function that outputs node $z$

**Output:** A tree $t$ consisting of component contributions $c$, such that components $\sum_{c \in t} c = z$. The structure of $t$ is a nested list where each list represents a level in the tree

  **function** REPDECOMPOSE($z$)

    **if** is_nonlinear($z.f$) **then**

      **return** $[z]$

    **else if** is_unary($z.f$) **then**                     ▷ *Function is unary linear*

      $z_0 \leftarrow z$.parents()

      $t_0 \leftarrow$ REPDECOMPOSE($z_0$)

      **if** is_reduction($z.f$) **then**

        $t_{0,u} \leftarrow$ unbind($t_0$)       ▷ *Unbinds each $c \in t$ along the reduction dimension*

        $f_d \leftarrow$ decomp($z.f$)       ▷ *Returns $f_d$ such that $\sum_{c \in t_{0,u}} f_d(c) = z.f(z_0)$*

        **return** map($f_d, t_{0,u}$)       ▷ *Maps each $c \in t_{0,u}$ to $f_d(c)$*

      **else**

        **return** map($z.f, t^0$)       ▷ *Maps each element $c \in t$ to $z.f(c)$*

    **else**                          ▷ *$z.f$ is binary*

      $z_0, z_1 \leftarrow z$.parents()       ▷ *Get the parents of $z$ in the graph (inputs to $z$.function)*

      $t_0, t_1 \leftarrow$ REPDECOMPOSE($z_0$), REPDECOMPOSE($z_1$)

      $f_{d,0}, f_{d,1} \leftarrow$ decomp_binary($z.f$)       ▷ *Returns $f_{d,0}, f_{d,1}$ such that:*

      **return** [map($f_{d,0}, t_0$), map($f_{d,1}, t_1$)]   ▷ *$\sum_{c \in t_0} f_{d,0}(c) + \sum_{c \in t_1} f_{d,1}(c) = z.f(z_0, z_1)$*

# D Stepwise breakdown of the operation of RepDecompose on a vanilla attention-MLP block

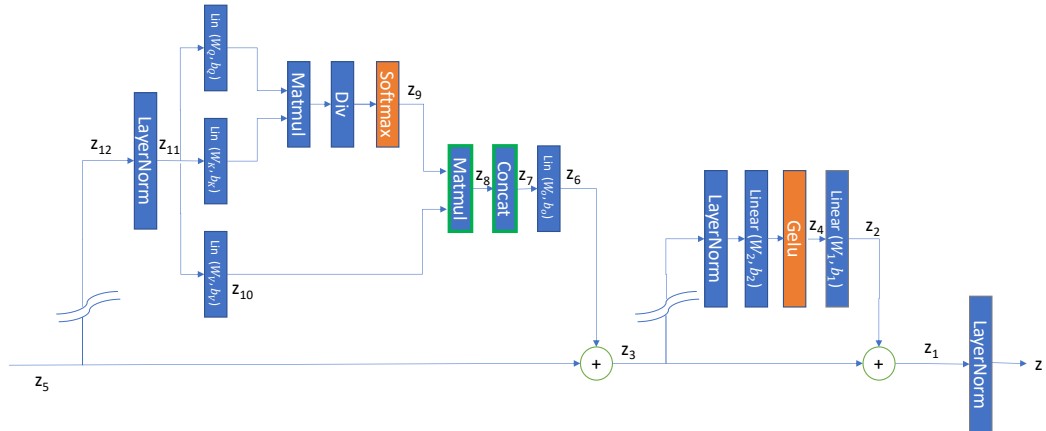

Figure 6: Illustration of a simple attention-mlp block. Intermediate tensors marked as $z_i$, non-linear nodes in orange, nodes where a tensor is reduced along a dimension of interest (tokens, attention head, etc) are marked by green borders

To illustrate the workings of our algorithm, we describe the steps that the RepDecompose algorithm takes on a simple attention-mlp block of a vanilla ViT transformer. Please refer to Figure 6 for the variable names in the following explanation.

First, we mark (with green borders in the figure) the computational nodes in which the contributions of the components get reduced. For the tokenwise contributions, this is the 'matmul' operation, while for the attention head contributions, it is the 'concat' operation. We also detach the graph at the input of each component to stop the algorithm from gathering only the direct contributions and not any higher-order contribution terms arising from the interaction between multiple components. Let the RepDecompose function be denoted by $d(.)$ which takes in a representation and returns an array of contributions. Here, $n$, wherever it appears, is the number of contributions in the decomposition of the input. The $\text{map}(f, d(z))$ operation applies $f$ to every contribution vector in $d(z)$. At each step, it is ensured that the sum of all contribution vectors/tensors in the RHS is equal to the vector/tensor that is being decomposed in the LHS via the distributive property for linear transformations. Then:

1. $d(z) = \text{map}(\lambda x. \frac{1}{\sigma}(x - \frac{\mu}{n}), d(z_1))$ (LayerNorm linearized as in Gandelsman et al [1], $n$ here is the number of contributions in $d(z_1)$)

2. $d(z_1) = (d(z_2), d(z_3))$

3. $d(z_2) = \text{map}(\lambda x. x W_1 + \frac{b_1}{n}, d(z_4))$ ($n$ here is the number of contributions in $d(z_4)$ )

4. $d(z_4) = [z_4]$ (stops when it hits a non-linear node)

5. $d(z_3) = (d(z_5), d(z_6))$

6. $d(z_6) = \text{map}(\lambda x. x W_o + \frac{b_o}{n}, d(z_7))$ ($n$ here is the number of contributions in $d(z_7)$ )

7. $d(z_7) = [[\text{zeropad}(v) \text{ for } v \in u] \text{ for } u \in d(z_8)]$ (Concatenation of a tensor along a dimension can be expressed as a sum of zero-padded tensors)

8. $d(z_8) = [[uv \text{ for } (u, v) \in \text{zip}(U.\text{cols}, V.\text{rows})] \text{ for } U \in d(z_9) \text{ for } V \in d(z_{10})]$ (via the distributive property for matrix multiplication)

9. $d(z_9) = [z_9]$ (stops when it hits a non-linear node)

10. $d(z_{10}) = \text{map}(\lambda x. x W_v + \frac{b_v}{n}, d(z_{11}))$ ($n$ here is the number of contributions in $d(z_{11})$ )

11. $d(z_{11}) = \text{map}(\lambda x. \frac{1}{\sigma}(x - \frac{\mu}{n}), d(z_{12}))$ ($n$ here is the number of contributions in $d(z_{12})$ )

12. $d(z_{12}) = [z_{12}] = [z_5]$ (Stopped since the comp graph is detached, if not the algorithm would return higher-order terms.)

# E    Proof of Theorem 1

*Proof.* From the first condition on **intra-component rank ordering**, for any two vectors $\boldsymbol{u}, \boldsymbol{v}$ and a linear map $f_i$, if $\|\boldsymbol{u}\| \leq \|\boldsymbol{v}\|$ then $\|f_i(\boldsymbol{u})\| \leq \|f_i(\boldsymbol{v})\|$. We first show that $f_i$ is a scalar multiple of an isometry.

If $\|\boldsymbol{u}\| = \|\boldsymbol{v}\| \neq 0$, then both $\|\boldsymbol{u}\| \leq \|\boldsymbol{v}\|$ and $\|\boldsymbol{v}\| \leq \|\boldsymbol{u}\|$. This implies that $\|f_i(\boldsymbol{u})\| \leq \|f_i(\boldsymbol{v})\|$ and $\|f_i(\boldsymbol{v})\| \leq \|f_i(\boldsymbol{u})\|$. Therefore, $\|f_i(\boldsymbol{u})\| = \|f_i \boldsymbol{v}\|$, when $\|\boldsymbol{u}\| = \|\boldsymbol{v}\|$. Given the input space of the transformation as $U$, we choose a unit vector $\boldsymbol{u}_{\text{unit}} \in U$. Let's assume $\|f_i(\boldsymbol{u}_{\text{unit}})\| = c$. With the above result, we can use the following equality $\|\frac{\boldsymbol{u}}{\|\boldsymbol{u}\|}\| = \|\boldsymbol{u}_{\text{unit}}\|$ to obtain the following:

$$\left\| \frac{f_i(\boldsymbol{u})}{\|\boldsymbol{u}\|} \right\| = \left\| f_i\left( \frac{\boldsymbol{u}}{\|\boldsymbol{u}\|} \right) \right\| = \|f_i(\boldsymbol{u}_{\text{unit}})\| = c, \tag{1}$$

Therefore:

$$\|f_i(\boldsymbol{u})\| = c\|\boldsymbol{u}\| \tag{2}$$

Thus, the linear transformation $f_i$ is a scalar multiple of an isometry. Now consider two linear maps $f_i$ and $f_j$ such that $\|f_i(\boldsymbol{u})\| = c_i\|\boldsymbol{u}\|$ and $\|f_j\boldsymbol{u}\| = c_j\|\boldsymbol{u}\|$. From the second condition on **inter-component rank ordering**, for any two vectors $\boldsymbol{u}, \boldsymbol{v}$ and linear maps $f_i, f_j$, if $\|\boldsymbol{u}\| \leq \|\boldsymbol{v}\|$ then $\|f_i(\boldsymbol{u})\| \leq \|f_j(\boldsymbol{v})\|$. This implies that if $\boldsymbol{u} = \boldsymbol{v}$, $\|f_i(\boldsymbol{u})\| = \|f_j(\boldsymbol{u})\|$. However, this can only happen when $\|f_i(\boldsymbol{u})\| = c\|\boldsymbol{u}\|$ for some constant $c$ for all $f_i \; \forall i$.

With this, let's denote $\frac{f_i}{c}$ as an isometry. One of the general property of isometries are that they preserve the inner product between two vectors $\boldsymbol{u}$ and $\boldsymbol{v}$. First we prove that isometries preserve the inner product, which we will then use to prove the orthogonality of $\frac{f_i}{c}$. Given two vectors $\boldsymbol{u}$ and $\boldsymbol{v}$, their inner product can be expressed as the following:

$$\boldsymbol{u}^T \boldsymbol{v} = \frac{1}{4}(\|\boldsymbol{u} + \boldsymbol{v}\|^2 + \|\boldsymbol{u} - \boldsymbol{v}\|^2) \tag{3}$$

An isometry by definition preserves the norm of the vectors i.e. $\|f_i(\boldsymbol{u})\| = \|\boldsymbol{u}\|$ and $\|f_i(\boldsymbol{v})\| = \|\boldsymbol{v}\|$. Due to this property, we can express the following relations:

$$\|f_i(\boldsymbol{u} + \boldsymbol{v})\| = \|\boldsymbol{u} + \boldsymbol{v}\|, \tag{4}$$

and

$$\|f_i(\boldsymbol{u} - \boldsymbol{v})\| = \|\boldsymbol{u} - \boldsymbol{v}\|, \tag{5}$$

We can express $f_i(\boldsymbol{u})^T f_i(\boldsymbol{v})$ as the reduction from Eq.(3):

$$f_i(\boldsymbol{u})^T f_i(\boldsymbol{v}) = \frac{1}{4}(\|f_i(\boldsymbol{u}) + f_i(\boldsymbol{v})\|^2 + \|f_i(\boldsymbol{u}) - f_i(\boldsymbol{v})\|^2), \tag{6}$$

$$f_i(\boldsymbol{u})^T f_i(\boldsymbol{v}) = \frac{1}{4}(\|f_i(\boldsymbol{u} + \boldsymbol{u})\|^2 + \|f_i(\boldsymbol{u} - \boldsymbol{v})\|^2), \tag{7}$$

Next we substitute the relations from Eq.(4) and Eq.(5) to Eq.(7) to obtain the following inner product preservation property:

$$f_i(\boldsymbol{u})^T f_i(\boldsymbol{v}) = \frac{1}{4}(\|\boldsymbol{u} + \boldsymbol{v}\|^2 + \|\boldsymbol{u} - \boldsymbol{v}\|^2) = \boldsymbol{u}^T \boldsymbol{v} \tag{8}$$

Next we use the inner product preservation property to prove the orthogonality of $\frac{f_i}{c}$ as follows:

$$f_i(\boldsymbol{u})^\top f_i(\boldsymbol{v}) = c^2 \boldsymbol{u}^\top \boldsymbol{v}, \tag{9}$$

$$\boldsymbol{u}^\top \left( \frac{1}{c^2} f_i^\top f_i - I \right) \boldsymbol{v} = 0, \tag{10}$$

From 10, we can infer the orthogonality of $\frac{f_i}{c}$ which leads to the following result:

$$f_i^\top f_i = c^2 I = kI, \tag{11}$$

$\square$

# F  Scoring function

---

**Algorithm 3** Scoring function for attributing properties to components

---

**Input:** $Z$, the image representation output by the model over $n$ images with dimension $d$ (shape: $n \times d$); $C$, the contribution of a particular component of interest (shape: $n \times d$); $B$, the set of $k$ feature vectors that represent a given feature (shape: $k \times d$)

**Output:** A score that signifies the importance of the component for the given feature

  **function** COMPATTRIBUTE($C$, $Z$, $B$)

     $B \leftarrow$ orthogonalize($B$)

     $s_Z \leftarrow ZB^\top$

     $s_C \leftarrow CB^\top$

     $r \leftarrow$ correlation_coefficient($s_Z$, $s_C$, dim=0)

     **return** mean($r$)

---

# G    Text-based Image retrieval

| Model name | Color | Texture | Animal | Person | Location | Pattern | Shape |
|---|---|---|---|---|---|---|---|
| DeiT | 0.679 | 0.563 | 0.774 | 0.596 | 0.818 | 0.597 | 0.764 |
| DINO | 0.663 | 0.657 | 0.781 | 0.742 | 0.833 | 0.680 | 0.706 |
| DINOv2 | 0.751 | 0.614 | 0.875 | 0.714 | 0.857 | 0.597 | 0.510 |
| SWIN | 0.795 | 0.720 | 0.904 | 0.780 | 0.912 | 0.760 | 0.739 |
| MaxVit | 0.872 | 0.832 | 0.911 | 0.828 | 0.901 | 0.803 | 0.797 |

Table 4: Spearman rank correlation for various common properties

# H Image-based Image retrieval

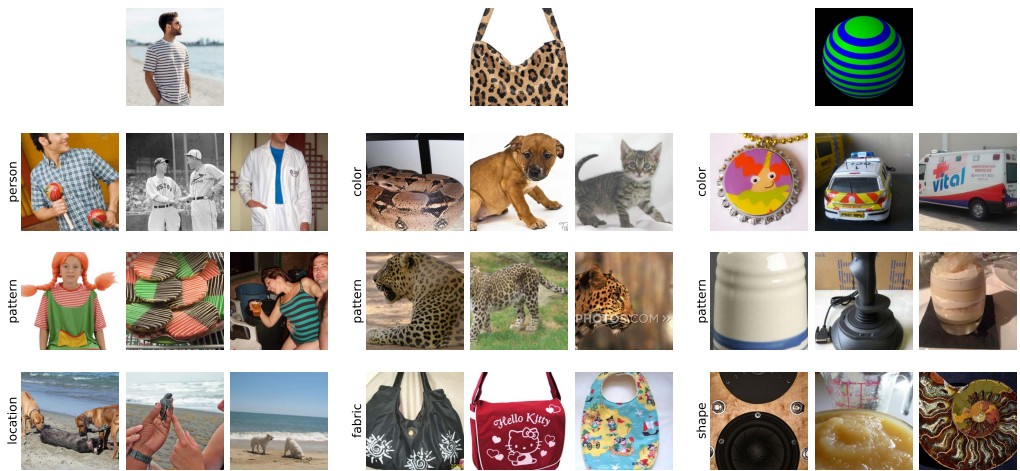

Figure 7: Top-3 images retrieved by the most significant components for various relevant properties for DINO

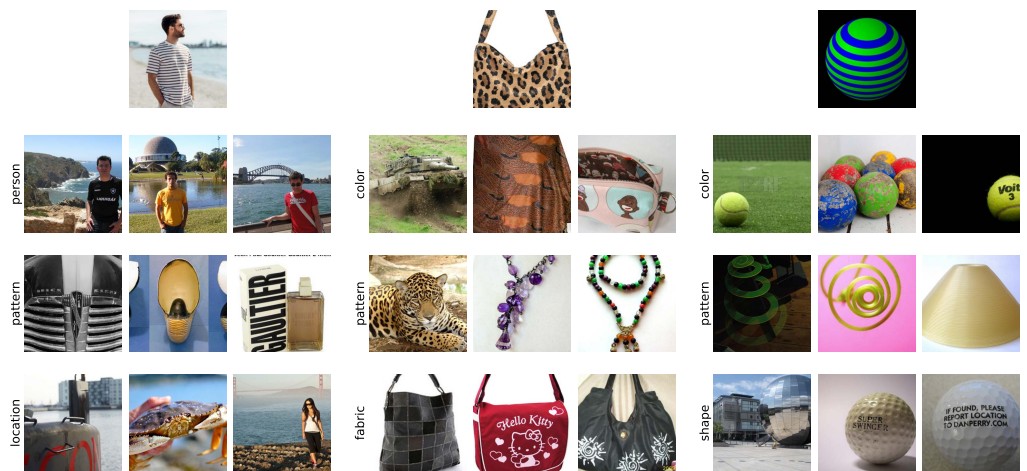

Figure 8: Top-3 images retrieved by the most significant components for various relevant properties for SWIN

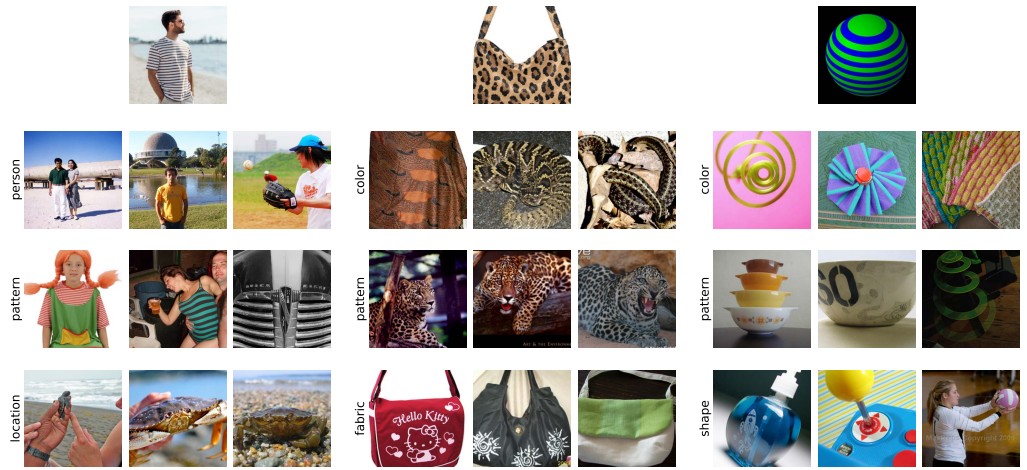

Figure 9: Top-3 images retrieved by the most significant components for various relevant properties for DeiT

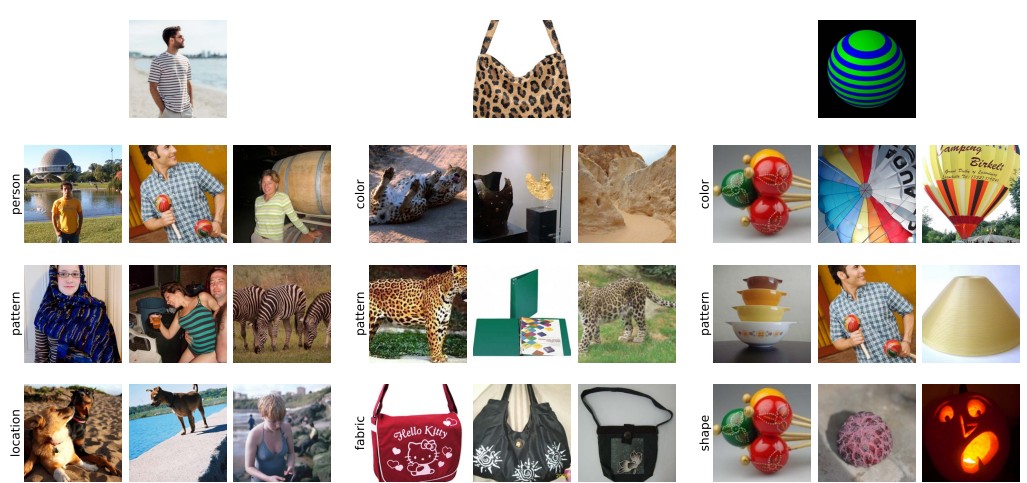

Figure 10: Top-3 images retrieved by the most significant components for various relevant properties for MaxViT

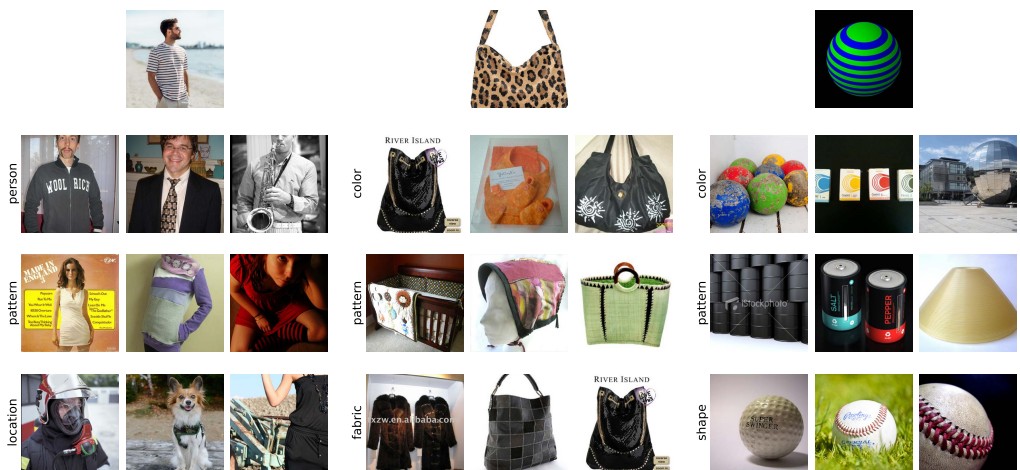

Figure 11: Top-3 images retrieved by the most significant components for various relevant properties for DINOv2

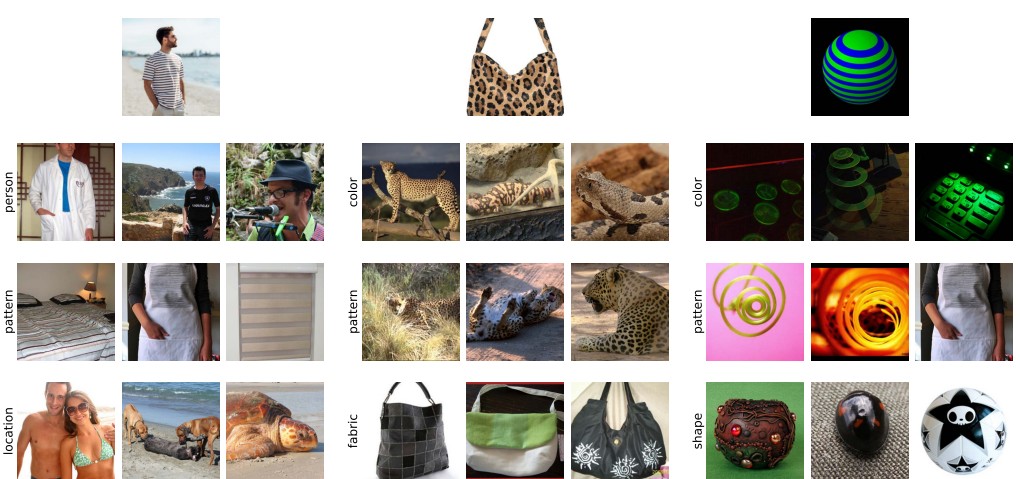

Figure 12: Top-3 images retrieved by the most significant components for various relevant properties for CLIP

# I Property visualization via token decomposition

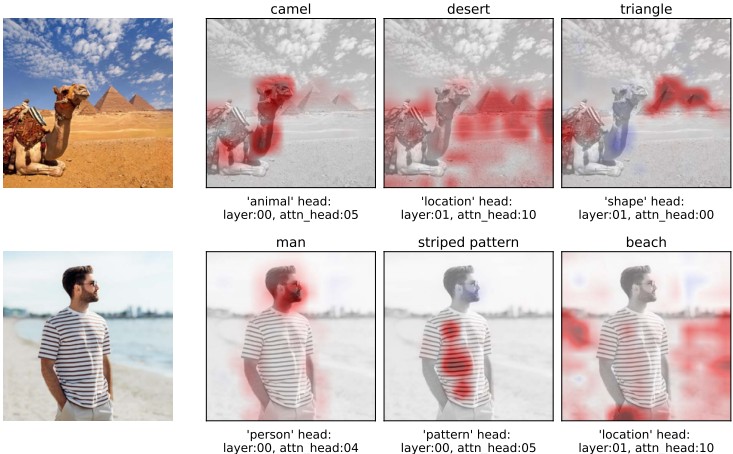

Figure 13: Visualization of token contributions for CLIP

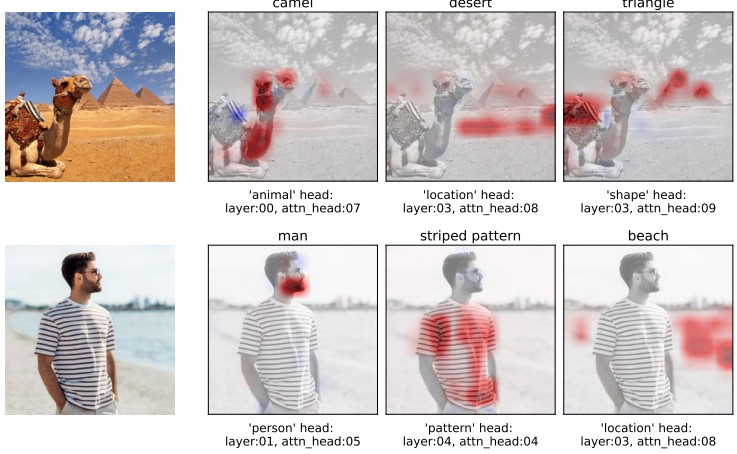

Figure 14: Visualization of token contributions for DINO

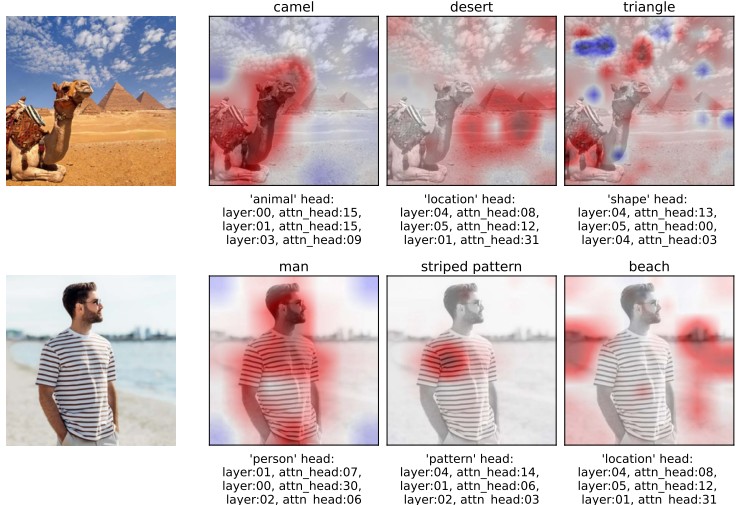

Figure 15: Visualization of token contributions for SWIN

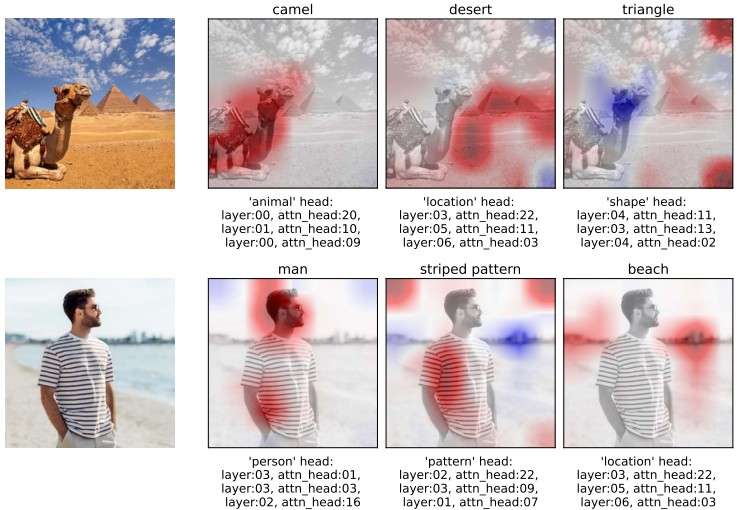

Figure 16: Visualization of token contributions for MaxVit

# J  Zero-shot segmentation

| Algorithm | DeiT | | | DINO | | | MaxVit | | | SWIN | | |
|---|---|---|---|---|---|---|---|---|---|---|---|---|
| | pixAcc | mIoU | mAP | pixAcc | mIoU | mAP | pixAcc | mIoU | mAP | pixAcc | mIoU | mAP |
| Decompose | **0.7719** | **0.5291** | **0.8305** | **0.7577** | **0.4863** | **0.8111** | **0.7163** | **0.4237** | **0.7237** | **0.7136** | **0.4338** | **0.7620** |
| Chefer et al. [8] | 0.7307 | 0.4785 | 0.7870 | 0.7309 | 0.4541 | 0.8080 | - | - | - | - | - | - |
| GradCam | 0.6533 | 0.4625 | 0.7129 | 0.7045 | 0.4309 | 0.7481 | 0.4732 | 0.1705 | 0.4243 | 0.5973 | 0.2360 | 0.5365 |

Table 5: Zero-shot segmentation results for different algorithms and models. Chefer at al 's code does not support MaxViT and SWIN models.

# K   Zero-shot spurious correlation mitigation

| Model name | Waterbird in water | Waterbird in land | Landbird in water | Landbird in land |
|---|---|---|---|---|
| DeiT | $0.985 \rightarrow 0.971$ | $0.733 \rightarrow 0.815$ | $0.787 \rightarrow 0.886$ | $0.991 \rightarrow 0.980$ |
| CLIP | $0.920 \rightarrow 0.814$ | $0.507 \rightarrow 0.746$ | $0.534 \rightarrow 0.744$ | $0.948 \rightarrow 0.857$ |
| DINO | $0.985 \rightarrow 0.944$ | $0.800 \rightarrow 0.911$ | $0.832 \rightarrow 0.943$ | $0.982 \rightarrow 0.956$ |
| DINOv2 | $0.994 \rightarrow 0.989$ | $0.967 \rightarrow 0.981$ | $0.971 \rightarrow 0.978$ | $1.000 \rightarrow 0.997$ |
| SWIN | $0.989 \rightarrow 0.989$ | $0.834 \rightarrow 0.871$ | $0.893 \rightarrow 0.923$ | $0.994 \rightarrow 0.994$ |
| MaxVit | $0.959 \rightarrow 0.942$ | $0.796 \rightarrow 0.814$ | $0.777 \rightarrow 0.832$ | $0.970 \rightarrow 0.961$ |

Table 6: All group accuracies on the Waterbirds dataset before and after component ablation

