# OpenReview forum: "Decomposing and Interpreting Image Representations via Text in ViTs Beyond CLIP"
_NeurIPS.cc/2024/Conference — NeurIPS 2024 poster_

### Official Review · Reviewer_HytW · 2024-06-22

**Soundness:** 1
**Presentation:** 2
**Contribution:** 2
**Rating:** 4
**Confidence:** 4

**Summary:**

The authors extend the idea proposed in [10] of decomposing CLIP’s image representation and interpreting the decomposed components via text, to models other than CLIP via learning a set of mapping functions, one for each decomposed component, to map the representation of a model to be analysed (e.g., DINO) to the CLIP space, such that they can perform text-retrieval on the decomposed component with the CLIP text encoder, and assign the component a textual label. They further automate the decomposition process by making use of the computation graph of the model, proposing an algorithm called RepDecompose to achieve this. This allows for a more flexible decomposition for transformer models, especially models which are not based on the plain ViT architecture. They also score the decomposed components according to their relevance with a text such that they can select several components for a given textual feature. Applications such as image-text retrieval, image-image retrieval, token importance visualization, and mitigating spurious correlations on Waterbirds dataset are shown.

**Strengths:**

- The authors are tackling a very important direction of interpreting models other than CLIP, with text. Text-based interpretation has lately (to some big extent) been limited to CLIP models only, due to their strong image-text alignment which allows for human-friendly textual interpretation. However, computer vision is absolutely not just about CLIP, and there are an abundance of different models that are not trained for alignment, and that are used everywhere: in the same and similar research areas, in other research areas, and in industry. Therefore, we are presented with a serious bottleneck in interpreting models which are not trained for alignment, via text. I myself as a person working in this field, have previously questioned the direction this paper is proposing. Therefore, I am happy that the authors took this direction, and I believe it will have a large impact on the community.

- Figure 2 in a nice analysis, and the outcomes are interesting. It is compelling to see the mean-ablation experiments across different models trained with different datasets and supervision.

- RepDecompose algorithm allows us to apply decomposition to a Transformer regardless of the type of attention module design (assuming RepDecompose is valid – more on this in the weaknesses).

**Weaknesses:**

I would like to congratulate the authors for their work. Unfortunately, there are flaws that should be addressed. I will take the time to explain those in detail:

Major Weaknesses:

[W1] In my opinion, the authors are just interpreting CLIP features rather than the other models such as DeIT, DINO…etc. The problem with learning to map features from any model such as DINO, to the CLIP space (e.g., via a linear layer here) essentially means interpreting what the CLIP vision encoder learns. I will take some time to explain this: First let’s simplify the problem: assume we have a ViT model (e.g., DINO model) we want to interpret, and the CLIP vision encoder. Given an image, we want to train a one-to-one mapping from the features of the last layer of DINO (v_dino) to the features of the final projected global representation of CLIP (v_clip), which serves as the ground-truth for the mapper. If we assume a perfectly learned mapping function which maps v_dino to v_clip, then what we get out of the mapper is essentially v_clip. This is equivalent to just running the image through the CLIP vision encoder and obtaining v_clip (the ground-truth) directly. This means that what is interpreted after the mapping is v_clip rather than v_dino, and conveys what CLIP learns rather than what DINO learns. This simplification extends to mapping a component from any model (e.g., DINO), denoted as c_i, to v_clip. The authors put no restriction on preserving the semantics of c_i. There are no experiments to proof that the mapped representation still encodes what the original model (e.g., DeIT, DINO) learns. Since this is the topic of the paper, the authors should have addressed this before proceeding with any other experiment. Moreover, the regularization component in Theorem 1 has no effect in preserving the component role, as it decreases the cosine score by only 0.01 according to Table 1, which means it has a negligible effect (here I assume decrease is better because we should assume that the transformed representation is in the CLIP space, but should also be different from CLIP features for the input image). Therefore, these linear mappers could learn anything such that the addition of their output becomes the original CLIP representation v_clip, and the original semantics of c_i is lost. In summary, there is no restriction to keep the original semantics of c_i when mapping it to CLIP space, which means the interpretation is for CLIP’s representation rather than the models beyond CLIP. This deviates away from the title of the paper.

[W2] The authors did not show the validity of RepDecompose algorithm. The authors should first obtain the representation of the components for some plain ViT model by applying the simple decomposition as in [10] which is simple to apply in case of a plain ViT. Then they should run RepDecompose on that same plain ViT, and show that indeed, the representations are the same and equivalent. The authors have tried to partially address this via comparing the match rate with that from [10], as shown in Table 1. It is shown that the match rate is extremely low (0.18), which disproves the validity of the algorithm.

Moderate Weaknesses:

[W3] In Figure 3, the authors show that as the component score decreases, the retrieved images grow less relevant. This should not be validated qualitatively only. The authors should validate this quantitatively, for example by reporting the Area-Under-Curve, where the x-axis represents the number of components being removed, and the y-axis representing the similarity score between the image and text. This should also be compared across the different models analysed, and also compared with baseline CLIP: Can the global image representation of the baseline CLIP (denoted as Z_CLIP by the authors) still do a good job in this? Or do we really need its components to do this task? Again, this should be shown quantitatively. Finally, the authors showed an experiment in Table 2 where they report the correlation coefficient between (Z_CLIP, text) and (sum of most highly-ranked components, text), but the problem here is that if the authors consider (Z_CLIP, text) as the ground-truth, this means that they assume Z_CLIP can perform this retrieval task. In that case, what is the point of retrieval using the component outputs of the model as the vision source? If I understand correctly, the authors would like to perform property-based retrieval to validate the effectiveness of the components and the features they are responsible for encoding. But setting the ground-truth as Z_CLIP does not validate this, because if we reach the ground-truth score, it means we can perfectly do the retrieval via Z_CLIP.

[W4] There are no quantitative experiments on visualizing token contributions. For example, the work of [10] conducts zero-shot segmentation experiments and compares with other methods, but the authors do not perform such analysis. How do the different models analysed perform on zero-shot segmentation? How do they compare with existing methods in the literature and to [10]?

[W5] The pseudo-algorithm in Page 4 is not clear and there are a lot of unexplained steps. Moreover, the authors mention they stop the recursion and traversal when they “hit a non-linear node”. The MLP module in the transformer involves running each token individually through the MLP, and therefore there is no linear operation in the MLP. such as summation. How do you traverse through this? This causes the algorithm to stop, and the parent nodes (here the attention heads) cannot be obtained since the traversal stops. What is meant by a binary z.f? In general, the algorithm should be understood to the point that readers are able implement it on their own (general idea is not enough – and listing the algorithm without explaining it is also not enough). I would recommend the authors to enhance the understanding via a diagram example for one attention-mlp block of the transformer, and explain clearly each step written in the algorithm. That is a very important part I was expecting to see in the supplementary material. Readability and a clear understanding is a vital part of any research work.

Minor Weaknesses:

- I do not see the scoring function as a contribution. In essence the scoring function is the dot product between the mapped representation f(c_i) and the encoded text concept (e.g., stripes), which comes for free when finding the component's matched text concept (since the text concept assigned to a component is the one with the highest similarity score).

- Line 93: The work of [10] clearly shows that the MLPs have no effect and are mean-ablated, and thus it should be mentioned that the MLP term is ablated (replaced by the mean value).

- Texts in Table 1 should be differed to other sections or to the supplementary material. These are qualitative examples rather than quantitative…. It is not a good practice to mix the two in one table.


In summary, the paper requires significant changes and additional experiments to be ready for NeurIPS. So my decision will be, sadly enough, to reject this paper.

**Questions:**

If the authors have a different opinion on W1 and W2, I would like to hear it, supported by experimental evidence.

I understand that rebuttal time is limited, so I encourage the authors to address as much as possible from W3-W4-W5.

The minor weaknesses are comments which do not affect my decision.

**Limitations:**

Limitations are addressed in the supplementary material. Considering that interpretability is very important for several critical applications, weakness W1 (if not addressed) would mislead users into incorrect interpretations.

---

> ### Author Rebuttal · Authors · 2024-08-03
>
> Thank you for your detailed comments on our paper, we appreciate the feedback. We are glad that you find the direction of our work important and interesting. However, we disagree with your characterization of our work on multiple points. We answer your concerns below:
>
> > [W1]  In my opinion, the authors are just interpreting CLIP features ...
>
> We disagree with this comment. It is true that by mapping to CLIP space and interpreting using the CLIP text encoder, we are "constrained" to using concepts that CLIP has learnt well. This does not mean, however, that this method of interpretation is ungrounded. Our results from the experiments on image-based image retrieval and zero-shot spurious correlation are evidence against this. In both these experiments, the mapping functions $f_i$ are only used for selecting the components to be used or ablated based on some specific property, and not used in performing the task itself (like image retrieval and inference on data with spurious correlations). If it were true that the property encoded in $f_i(c_i)$ is different from $c_i$, we would not observe property relevant images being retrieved or any significant improvement in worst group accuracy.
>
> > The authors put no restriction on preserving the semantics of c_i. ..
>
> We discuss this in Section 4 Theorem 1 and the following paragraph. We show that an orthogonal map preserves the relative importance of features, since it is an isometry. We encourage $f_i$ to be orthogonal during training.
>
> >  the regularization component in Theorem 1 has no effect in preserving the component role ....
>
> The cosine distance (the first term in the loss function) measures how well  $\sum f_i(c_i)$ match z_CLIP. The orthogonality regularizer (the second term) encourages the mapping to be an isometry so as to not distort the relative features (like you mentioned). Since these are two distinct objectives, the regularization component is not expected to have any effect on the cosine similarity - in fact it is a bit surprising that the cosine distance marginally decreases. The relevant metric for faithfulness of the mapping is the match rate which measures how well the TextSpan descriptions of a component (using the ImageNet trained linear head as the "text bank") match the corresponding TextSpan description of the *same* component after CLIP mapping and using the CLIP text embeddings of ImageNet classes as text bank. We observe a significant improvement in the match rate when the regularizer is introduced (in Table 1).
>
> However, you indeed have a point that the improvements are not very large. This is because any randomly sampled high dimensional matrix is approximately orthogonal already, as for any two independent zero-mean Gaussian/uniform random variables $x, y$,  $E[xy] = 0$ . In our experiments, training this matrix via gradient descent does not significantly change this property. However, since the orthogonality property is crucial to preserve the faithfulness of the mapping, we explicitly add it in the loss term. With this, the norm $\|| f_i f_i^T - I \||_F$ decreases by around 50% and match rate increases by around 0.03.
>
> > [W2] The authors did not show the validity of RepDecompose algorithm. ...  It is shown that the match rate is extremely low (0.18), which disproves the validity of the algorithm.
>
> > [W5]  The pseudo-algorithm in Page 4 is not clear and there are a lot of unexplained steps.
>
> We address W2 and W5 together. RepDecompose is a generalization of the manual process by which Gandelsman et al computed component contributions. We have added a step-by-step explanation of the execution of RepDecompose on an attention-mlp block in the global rebuttal. This is still a simplified graph for illustrative purposes, the actual computational graph has many more nodes and is somewhat more complicated. We provide the code we use in the supplementary, the RepDecompose algorithm can be found in the file `helpers/linear_decompose.py`.
>
> The match rate in Table 1 (as mentioned previously) is only meant to validate the faithfulness of the mapping functions $f_i$, and it is intended as a metric for CompAlign. It does not validate the RepDecompose algorithm in itself. The contributions returned by RepDecompose match those computed by Gandelsman et al exactly in the case of vanilla ViTs.
>
> We acknowledge that the pseudocode in Algorithm 1 is unclear. While the intuition is very simple ( traverse through linear nodes by applying the distributive property and stop at non-linear nodes), the actual implementation is lengthy and requires extensive handling of edge cases which might have obscured the intuition. We will present the pseudocode at a higher level of abstraction in the camera ready.
>
> > [W3]
>
> Our experiments on text-based image retrieval using components is only meant to show that (a) identifying certain image properties can be localised to certain individual components, and (b) our scoring function provides a useful signal for identifying these components. We do not claim that the individual components can retrieve images better than the overall representation or the CLIP representation.  We show this quantitatively in Table 2 and 4. Please let us know If there are any concerns with the experimental setup for this section.
>
> > [W4]  ... quantitative experiments on visualizing token contributions ...
>
> Thank you for your suggestion. We have now added results of our experiments on image segmentation with competitive zero-shot segmentation baselines like GradCam and Chefer et al's method in the global rebuttal. Some of the methods are not equipped to work with certain models, in which case the cells have been left blank.
>
> We hope that we were able to sufficiently address your concerns in the rebuttal. Please let us know about any remaining issues or concerns, and do consider increasing your score if you are satisfied with our answers.

---

> ### Comment · Reviewer_HytW · 2024-08-09
>
> I thank the authors for addressing my concerns. However, I still think W1 is not addressed. I specifically asked for experimental evidence (other than the match rate) for W1 in the rebuttal, which the authors did not provide. The match rate is extremely low: 0.185, and improves over the unfaithful baseline by only 0.03. Finally, having one metric to evaluate a whole new direction and the faithfulness of the method, is not convincing to me. The authors could formulate 1-2 experiments to prove faithfulness and that the semantics of the original model, is still preserved. I am also happy to provide suggestions on how to evaluate this, should the authors want to.
>
> I am raising my rating by one score, but I cannot give any form of accept for this paper, as this paper does not really satisfy it's objective as written in the title, and the authors have not addressed W1. In fact, the paper makes the problem worse because it leads to interpretation of a different model (e.g., it interprets DINO as CLIP). Although all reviewers suggest accept, i think i have a valid point and I stand by it.
>
> However, I will ensure to have fruitful discussions with the area chair and other reviewers after the rebuttal, to ensure that my decision is fair.
>
> In the meantime, if the authors would like to discuss on this matter, I would be happy to do so.

---

> ### Author Response · Authors · 2024-08-09
> **Regarding concerns with W1**
>
> Thank you for reading our rebuttal and updating your score. We are eager to address your remaining concerns with W1. But as stated in our rebuttal, we already offer strong experimental evidence in the form of **zero-shot spurious correlation mitigation** and **image-based image retrieval**. A quick recap:
>
> 1. We use RepDecompose to extract the direct contributions of these components $c_i$ such that the final representation $z = \sum c_i$
> 2. We use the trained linear maps $f_i$ from CompAlign to map each component $c_i$ to $f_i(c_i)$, and then using our scoring function, identify some salient properties encoded by these components
> 3. We then either select or ablate the top-k component contributions $c_i$ (and **not** $f_i(c_i)$) depending on the task, and perform image-based image retrieval or spurious correlation mitigation using these $c_i$.
>
> This shows that regardless of the "low match rate", our method as a whole is successful at identifying roles performed by the components so that they can be used to perform these tasks. Please not that **only** the $c_i$ are being used for these tasks. If the $f_i$ maps were not faithful to the "true" interpretation of the model components, we could not perform well on these tasks. We would like to understand the reason why you don't believe in these experiments.
>
> Also a note on match rate - we introduce this metric so that we can cleanly identify the delta between using the ImageNet pretrained heads vs CLIP text embeddings for interpretations. However, this metric has some flaws:
>
> 1. The match rate shown in the table is the exact match rate, and the approximate matches are not included in the metric.
> 2. CLIP text embeddings of the ImageNet classes are significantly different from the embeddings from the ImageNet trained fully conncected layer and may in fact represent different concepts. For example, CLIP's concept of a "library" may be different from the library images in ImageNet.
> 3. Some contributions may not have any clear interpretation, and in this case CompAlign is unfairly penalized since we take the ImageNet derived descriptions as proxy for ground truth.
>
> However, we observed that any *relative* improvements in this metric did correspond strongly to the quality of the visualizations. Our purpose in providing the metrics was to show the relative improvement of CompAlign as compared to the "one map only" baseline that can be found in prior literature. Prior work such as Moayeri et al [1] have already used a single linear map to interpret the final layer of vision models using CLIP. Note that we improve on this baseline by around 0.1, which is significant.
>
> We are happy to add any additional experiments that you suggest, if it can help in clarifying this point further.
>
>
> [1] Moayeri, M., Rezaei, K., Sanjabi, M. &amp; Feizi, S.. (2023). Text-To-Concept (and Back) via Cross-Model Alignment. *Proceedings of the 40th International Conference on Machine Learning</i>, in <i>Proceedings of Machine Learning Research* 202:25037-25060 Available from https://proceedings.mlr.press/v202/moayeri23a.html.

---

> ### Author Response · Authors · 2024-08-10
>
> Unfortunately there was an accidental omission in the reader list, which may have caused you to not see our response. We have edited the reader list now.

---

> ### Comment · Reviewer_HytW · 2024-08-11
>
> Thank you for the authors response. I would also like to apologize for the late reply.
>
> > we already offer strong experimental evidence
>
> These are not indicative of faithfulness. Let's take the image-based image retrieval as an example. These experiments are based on selecting $k$ components, mapping them, and adding the results up to form a visual feature representation in CLIP space. The authors mention in the supplementary material: "the number of components k used for the image-based image retrieval experiment was tuned on an image-by-image basis", and they do not mention what $k$ was used for each image. $k$ could be tuned for each image seperately such that the results are good. If $k$ is good enough, it would be enough to approximate the visual feature representation in CLIP space which at the end, is almost equivalent to using CLIP to encode the image. We don't even know what the individual mappers are learning, and now we come to the problem of "interpreting an interpretation technique". There could be many cases one out of an endless possibility of cases. There is no experiment showing what these mapper components are really doing and how faithful they are, which is what I asked for in my first review. Once this visual feature representation is achieved by the mapper output, it is already in CLIP space, and it is reasonable (and not surprising at all) to expect a retrieved image which matches the reference image.
>
> Throughout the rebuttal, it appears that the authors are avoiding conducting such experiments and instead are attempting to justify their work using existing experiments in the paper (and especially the match rate). In my previous response, I also gave the opportunity to the authors to select any other experiments to prove faithfulness. Since they did not, I could suggest two experiments (I didn’t give this a deep thought, these are from the top of my head), taking DINO as an example:
>
> - Take the image samples that were classified differently by DINO (using the trained linear head) and CLIP (using zero-shot classification). Let’s denote those samples by Y. Avoid all samples that were classified correctly by both DINO AND CLIP. Now, what is the accuracy of Y for DINO before and after? If the mappers really preserve the semantics of DINO, then the accuracy of Y for DINO should not change much.
> - The authors could use some interpretation technique such as Grad-CAM or Rollout or any other technique. This highlights relevant features. The authors could do this for the DINO model before and after the mapping. By then applying the Insertion and Deletion evaluation metrics for example (I assume the authors know them well if they work on interpretability), the Area-Under-Curve (AUC) should remain roughly the same.
>
> Again, I kindly ask the authors to avoid justifying their answers with already existing experiments or metrics in the paper. I am aware of the authors paper well, and have read it three times already, two times during the initial review, and one time during the rebuttal. My comments are meant to direct what I find missing and important in the authors work.

---

> ### Author Response · Authors · 2024-08-11
>
> > These experiments are based on selecting  components, mapping them, and adding the results up to form a visual feature representation in **CLIP space**.
>
> This is exactly the misconception that we wanted to address. The image-based image retrieval **does not involve** adding the mapped component contributions in **CLIP space**, but instead adds the component contributions **prior to the mapping**, which are still in **DINO space**  (or any other model space). The maps and the mapped components are **only** used in the *interpretation* process, and **not** during the *validation* of the interpretation. Thus, no matter what component contributions we add, they would always be in the representation space of the original model and not the CLIP space. The visual feature representation which is obtained by summing up the contributions is also in representation space of the original model.
>
> The same also holds true for the spurious correlation mitigation experiments. This is precisely why we specifically pointed out these two experiments as evidence, as these **do not involve** the mapping during CompAlign.
>
> >  The authors mention in the supplementary material: "the number of components k used for the image-based image retrieval experiment was tuned on an image-by-image basis", and they do not mention what  was used for each image.
>
> We mention in the next sentence that "$k$ is approximately around 9 for larger models like Swin or MaxVit, and around 3 for the rest." To be more precise, what we mean is we tune $k$ within a range of $9 \pm 4$ for SWIN/MaxVit and $3 \pm 2$ for the rest of the models. The main reason for this variation in $k$ is that we break down SWIN/MaxVit into many more components as compared to the rest, which means that each contribution vector for SWIN/MaxViT is less informative. We then select the top $k$ contributions strictly according to the scoring function. This is however beside the point because as we explained no matter what $k$ we choose, we can never recover the CLIP representation using the component contributions because **they are not in the CLIP space in the first place**. In addition, we also present spurious correlation mitigation experiments in which we do not tune $k$ at all.
>
> > Further experiments
>
> We will think more on the lines of the experiments you suggested and hope to add some results if time permits. However, we note that as currently formulated, the experiments do not cleanly disentangle the use of the mapping functions for interpretation and application. Ideally, there should be an "interpretation" stage where we use the maps to get some information about the models, and a "validation" stage where we use this information to manipulate the model somehow, without involving the maps or the mapped contributions in any form. Therefore, we still believe that the existing experiments are much more convincing.

---

### Official Review · Reviewer_FTxd · 2024-07-03

**Soundness:** 4
**Presentation:** 4
**Contribution:** 3
**Rating:** 6
**Confidence:** 5

**Summary:**

This paper introduces a framework for identifying the roles of various components in Vision Transformers (ViTs), extending beyond CLIP models. The framework consists of two key components: RepDecompose, which automates the decomposition of representations into contributions, and CompAlign, which maps these contributions into text space for intuitive interpretation. Additionally, the authors propose a novel scoring function to rank component importance. The framework's efficacy is demonstrated through applications in vision-language retrieval and mitigating spurious correlations in a zero-shot manner.

**Strengths:**

[**Writing**] The paper exhibits exceptional clarity and coherence in presenting its concepts. The accompanying figures serve as effective visual aids, significantly enhancing the comprehensibility of the discussed ideas.

[**Methodology**] The proposed method offers researchers a valuable tool for conducting in-depth analyses of vision transformers, providing clearer insights into their internal mechanisms. This unified analysis framework represents a significant step forward in enhancing the interpretability of complex, black-box models.

[**Empirical Evaluation**] The inclusion of comprehensive ablation experiments is a notable strength of this paper. These studies collectively offer a clear demonstration of the proposed algorithm's capabilities in interpreting the roles of different components within ViT models, providing crucial insights into model behavior and design choices.

**Weaknesses:**

[**Ambiguity in Introduction**] The description of insights from previous work [10] in line 22 lacks clarity. The statement "a sum of vectors over layers, attention heads, and tokens, along with contributions from MLPs and the CLS token" is ambiguous, as these components might have different dimensions and cannot be simply added. This lack of clarity compromises the self-contained nature of the manuscript, necessitating readers to refer to external sources for full comprehension.

[**Generalizability**] While the authors present RepDecompose as a general algorithm for ViT decomposition and analysis, the additional assumptions outlined in Section 3.1 raise questions about its broader applicability. For instance, (1) The concept of "direct" contributions implies that the model must incorporate residual connections for the algorithm to consider earlier layers, (2) The linearity assumption restricts the analysis to linear components only, and (3) The reduction of c_{i,t} to c_{i} through direct summation relies on a strong underlying assumption as well. These constraints may limit the algorithm's applicability across diverse ViT architectures

[**Limited Practical Applications**] While the paper makes strides in model understanding, it falls short in demonstrating practical applications of these insights. The current experiments, such as image retrieval and token visualization, primarily serve to validate the understanding process and can be considered preliminary downstream tasks. The manuscript would benefit from ablation studies that demonstrate how manipulating or removing specific components based on RepDecompose and CompAlign observations can enhance or reduce certain model capabilities. Such experiments would more convincingly illustrate the practical utility of the proposed framework in model improvement and optimization.

[**Structural and Writing Issues**]
* Figure 1 is never referenced in the main text.
* Line 14: Missing space between "features." and “These”

**Questions:**

All following questions are related to weakness points.
1. Given the assumptions presented in Section 3.1, how generalizable is the RepDecompose algorithm to ViT architectures that may not strictly adhere to these assumptions?
2. While image retrieval and token visualization provide insights into model behavior, have you considered more advanced downstream tasks that could demonstrate the practical benefits of your framework?
3. Have you conducted experiments to show how manipulating or removing specific components based on your RepDecompose and CompAlign observations affects model performance on specific tasks?
4. How might your framework be extended to not only understand but also guide the improvement of ViT models? Are there plans for future work in this direction?

**Limitations:**

The discussion on limitations about the paper is relatively thorough. The points raised in the weaknesses section further articulate limitations that should be considered. There is NO discussion on societal impact.

---

> ### Author Rebuttal · Authors · 2024-08-03
>
> Thank you for your favorable review. We are happy to hear that you found our work clear and coherent, with comprehensive evaluation. We answer your questions below:
>
> > Ambiguity in Introduction
>
> Thank you for the feedback. We will take care to expand and rephrase the introduction to remove this ambiguity. We have also added a step-by-step explanation of the execution of RepDecompose on one attention-mlp block in the global rebuttal. The reason why we can add multiple contributions together is because RepDecompose automatically applies the transformations to the output of each component ($z_i$ in the figure) so that the contributions $c_i$ are in the same vector space as $z$.
>
> Please let us know if there are any other points in the introduction that were unclear or ambiguous.
>
> > Generalizability
>
> It is true that for our algorithm to work, all non-linear components must be short-circuited via a residual/skip connection in the model architecture. Fortunately, this is almost always the case in modern neural networks, even in modern CNN architectures like ConvNext. This is because the skip connections are added to ensure that any "dead neurons" in the non linearities does not affect gradient flow. This means that it is now a safe assumption that non-linearities can be "skipped" using residual connections. It is also true that only the linear parts of the model can be decomposed, but modern ViTs are overwhelmingly linear in nature with very few non-linearities in practice. Therefore, we expect our method to generalize well across multiple modern architectures.
>
> > The reduction of c_{i,t} to c_{i} through direct summation relies on a strong underlying assumption
>
> The "underlying assumption" being referred to here is not very clear to us. Since the attention is linear in the OV circuit, it can be further decomposed as a sum over contributions from tokens. This holds as long as there are no non linearities in the OV circuit. Please do clarify further if we misunderstood you.
>
> > Limited Practical Applications
>
> This is a relevant point, thank you for bring this up. In our paper, we demonstrate a practical application for mitigating spurious correlations by ablating the output of a specific set of attention heads, identified using our method. You can refer to Section 6.4, where we show improvements in both worst-case and average group accuracy (see Table 3) for the Waterbirds dataset. This result illustrates that manipulating or removing specific components, as guided by our framework, can enhance model robustness. We have also added results on zero-shot segmentation comparing it with other competitive baselines such as GradCAM and Chefer et al's method. Our method outperforms the baselines whenever they are applicable for a given model.
>
> Our interpretability framework is primarily validated through downstream applications in text-based and image-based image retrieval, as demonstrated in Sections 6.1 and 6.2. Our findings however suggest that practical tasks like image-conditioned image retrieval based on specific features (e.g., patterns or locations) can potentially be performed in a zero-shot manner using any vision encoder, rather than requiring a specialized model. While achieving state-of-the-art results in this task is not the primary focus of our work, we believe our results lay the groundwork for unlocking various capabilities in vision encoders, all without the need for additional training.
>
> > Structural and Writing Issues
>
> We thank the reviewer for pointing out these issues. We will correct them in the final version of the paper.
>
> > Have you conducted experiments to show how manipulating or removing specific components based on your RepDecompose and CompAlign observations affects model performance on specific tasks?
>
> Our experiments on mitigating spurious correlations show that removing specific components helps in mitigating spurious correlations. We expect that our framework can also be applied to do model unlearning in an efficient manner by finetuning or removing specific components, however we leave this for future work.
>
> > While image retrieval and token visualization provide insights into model behavior, have you considered more advanced downstream tasks that could demonstrate the practical benefits of your framework?
>
> > How might your framework be extended to not only understand but also guide the improvement of ViT models? Are there plans for future work in this direction?
>
> Thank you for raising this important question. Beyond model understanding, in our paper we demonstrated the potential applications of our framework in spurious correlation mitigation, image/text-based image retrieval, and zero-shot segmentation. We believe our framework can also be applied to address issues related to harmful biases or copyrighted image information within the network. By using probe text descriptions, it is possible to identify model components that encode such biases or infringement information, which can then be ablated to develop safer vision models in a post-hoc way. However, we note that these applications, while impactful, are beyond the scope of the current paper and represent promising directions for future research.

---

> ### Comment · Reviewer_FTxd · 2024-08-08
> **Thanks for authors' response**
>
> Thank the authors for your response. The rebuttal clears most my previous concerns. Hence, I have decided to keep the original positive rating.

---

### Official Review · Reviewer_N9ji · 2024-07-10

**Soundness:** 3
**Presentation:** 2
**Contribution:** 3
**Rating:** 5
**Confidence:** 3

**Summary:**

This paper proposed a novel representation decomposition method for general ViT models. Then, with aligning the component representations to CLIP space, the decomposed contribution vectors can be interpreted through text using CLIP text encoder. Moreover, a scoring function is also proposed to assign an importance score to each component-feature pair. Multiple different types of ViT are analyzed.

**Strengths:**

1. This work proposes a general method to analyze various ViT models by decomposing the final representation into interpretable components.
2. Multiple applications are conducted and also demonstrate that the decomposition and interpretation are effective, such as image retrieval and visualizing token contributions

**Weaknesses:**

1. Lack of qualitative or quantitative comparison with the related works, such as the previous image representation decomposition method mentioned in Sec.3.
2. As for the applications of the decomposed components, there is a lack of a quantitative evaluation of the text or image-based image retrieval performance.

**Questions:**

1. Could the flowchart of REPDECOMPOSE and COMPALIGN also illustrated by a more intuitive figure? The algorithm in Sec3.1 and Sec4 and the corresponding descriptions are not very clear.
2. How to explain that with some models, the top-3 images retrieved by the most significant components for a specific property are unrelated to that property? For example, in Figure 10, with DINOv2 and “color” as the property, the retrieved images are just similar to the object in the target image, while not including similar color things.

**Limitations:**

Results are mostly exhibited by limited visualizations and a lack of strong evaluation metrics to support the effectiveness.

---

> ### Author Rebuttal · Authors · 2024-08-03
>
> Thank you for the favorable review. We answer your questions below:
>
> > Lack of qualitative or quantitative comparison with the related works, such as the previous image representation decomposition method mentioned in Sec.3.
>
> We would like to point out that our work can be viewed as an extension or generalization of the work from Gandelsman et al, which was specific to CLIP models. Our method, on the other hand, works on models with various architectures trained with different methods.
>
> We now include a comparison of our method with competitive baselines like Gradcam and Chefer et al's method in the global rebuttal. Note that some of the baselines are not equipped to work on architectures like SWIN or MaxVit, in which case we leave it blank.
>
> > lack of a quantitative evaluation of the text or image-based image retrieval performance.
>
> We have provided a quantitative evaluation of the text based image retrieval performance in Table 2 and 4 in the paper. In the paper, this is meant as a validation of our overall method as we compare the ability of various components to retrieve images wrt a certain property as measured by the CLIP score ranking.
>
> > Could the flowchart of REPDECOMPOSE and COMPALIGN also illustrated by a more intuitive figure? The algorithm in Sec3.1 and Sec4 and the corresponding descriptions are not very clear.
>
> We have added a step-by-step explanation of the execution of RepDecompose on one attention-mlp block in the global rebuttal, and we will add this in the camera-ready version should the paper be accepted. We will also revisit the CompAlign section and elaborate the algorithm further. In essence, the purpose of CompAlign is to map (or "align") the contributions to the shared CLIP space via an approximately orthogonal map.
>
> > How to explain that with some models, the top-3 images retrieved by the most significant components for a specific property are unrelated to that property?
>
> It is possible that in some cases, we do not find any component which predominantly encode the property of interest. In these cases, retrieving images based on these components may also return images which are similar in dimensions other than the specified property.
>
> We hope that your concerns with the paper are addressed with this, if there are still some remaining issue please do let us know. We request you to consider increasing the score if you are satisfied with our answers.

---

> ### Comment · Reviewer_N9ji · 2024-08-12
> **Response to rebuttal by authors**
>
> Thanks for the response. It addressed my concerns. I decide to keep my rating.

---

### Official Review · Reviewer_oPeN · 2024-07-11

**Soundness:** 3
**Presentation:** 3
**Contribution:** 3
**Rating:** 6
**Confidence:** 5

**Summary:**

The paper analyzes the direct contributions of individual layers and attention heads in vision models. Based on a similar idea to Gandelsman at al. that was applied to CLIP, this method decomposes the final representations of other models into individual contributions. To interpret these contributions, the paper proposes a method that translates the representations into the CLIP joint space.
The interpretations introduce additional application - spurious cues reduction, visualizing token contributions analysis and image-based image retrieval.

**Strengths:**

The paper is very clear and intuitive for readers who are familiar with TextSpan.

The presented matching algorithm and scoring algorithms are novel and allow us to interpret the hidden state of new vision models.

 The spurious cue reduction method is convincing and shows that the interpretation of the different components is grounded.

 There is a convincing qualitative and quantitative ablation study for COMPALIGN and the need for lambda.

I believe this paper presents a new opportunity for research into the mechanistic interpretability of non-CLIP models. I will recommend its acceptance if my concerns will be addressed.

**Weaknesses:**

A brief explanation of the TextSpan algorithm can be useful to understand the paper without the need for reading Gandelsman et al.

It is not clear that the presented scoring function is more useful than using TextSpan - a qualitative and quantitative comparison between the two approaches can provide a more convincing case for the introduction of this approach.

While the presented approach uses CLIP, it doesn't use the main advantage of this model - the fact that full complex sentences can be encoded to the same joint image-text space. The current experiments use very limited sets of descriptions for scoring and interpreting the individual components. The paper will greatly benefit from more qualitative and quantitative results that make use of large-scale text sets that include image descriptions (just like the ones presented in Gandelsman et al.). As the scoring algorithm can use these datasets instead of the small set of concepts, more text descriptions might introduce a new understanding of the different components.

Low match rate - it's not clear what is the reason behind the low match rate. One possibility is that CLIP is not expressive enough. It will be useful to show how does the match rate behaves for different CLIP models.

It is not clear why the representations should be mapped to CLIP image features first, as opposed to directly mapping them to a sparse linear combination of words in CLIP (i.e. via sparse autoencoder). An comparison to this approach can be very useful.

**Questions:**

Are individual heads spatialized mostly on specific tasks (e.g. counting/textures/colors) as shown in Gandelsman et al.?

What is the effect of using different CLIP models when doing the COMPALIGN? Is there any trend about how interpretable the models are,  given larger CLIP models?

What are the top text descriptions found by TextSpan/Scoring function for each mapped head? The paper will benefit from significantly more qualitative results.

**Limitations:**

The limitations of the paper are discussed.

---

> ### Author Rebuttal · Authors · 2024-08-03
>
> Thank you for your favorable review. We are glad that you found our work novel, clear and convincing. We address your questions below:
>
> > A brief explanation of the TextSpan algorithm can be useful to understand the paper without the need for reading Gandelsman et al.
>
> We agree and apologize for the oversight. We will add this in the appendix in the final version of the paper.
>
> > It is not clear that the presented scoring function is more useful than using TextSpan - a qualitative and quantitative comparison between the two approaches can provide a more convincing case for the introduction of this approach.
>
> Thank you for this point. We do not claim that our scoring function is superior to TextSpan - the motivation and use cases of the two are somewhat different. TextSpan is intended to automatically discover the roles of various components by mining text descriptions from a text bank. The text descriptions thus obtained are sometimes coherent and point to a well-defined role for the component, but at other times all over the place, making the attribution of a role to the component difficult (see the appendix of Gandelsman et al). Thus, TextSpan is a more unsupervised method which tries to understand the model's components on its own terms.
>
> Our scoring function, on the other hand, is useful for identifying components which have a role in representing some externally specified property. Since the property is specified manually, it is unlikely that there will be a specific component which corresponds to this property. This is the reason we may need to select multiple components corresponding to a particular property. Thus, we need a way to sort and select components which encode a given property, which is where our scoring function comes into picture.
>
> Leaving the motivation aside, TextSpan and our scoring function are closely related in a technical sense. Both use the variation of the component contributions with respect to a subspace determined by the text encoder. We also use TextSpan to validate CompAlign's mapping functions in Table 1.
>
> > While the presented approach uses CLIP, it doesn't use the main advantage of this model ... As the scoring algorithm can use these datasets instead of the small set of concepts, more text descriptions might introduce a new understanding of the different components.
>
> This is an interesting idea. We do not know if ImageNet trained/ self supervised models can learn to encode complex object relations (as present in "the dog chases after the cat") in the absence of text supervision. In fact, even CLIP models suffer from limited understanding of compositionality and are prone to treat the text prompt as a bag of words. However, we would have to defer this for future work owing to lack of space.
>
> > Low match rate - it's not clear what is the reason behind the low match rate. One possibility is that CLIP is not expressive enough. It will be useful to show how does the match rate behaves for different CLIP models.
>
> There may be multiple reasons for this:
> 1. The match rate shown in the table is the exact match rate, and the approximate matches are not included in the metric.
> 2. CLIP text embeddings of the ImageNet classes are significantly different from the embeddings from the ImageNet trained fully conncected layer.
>
> Note that even with the low match rate, we are able to determine components which perform a specific role well enough to do zero shot spurious correlation mitigation and image-based image retrieval.
>
> We tried two CLIP models, ViT-B-16 and ViT-L-14, as the source of $z_{\text{CLIP}}$ for the experiments and we found that the bigger model (ViT-L-14) yielded better cosine similarity and match rate. We would need to conduct more experiments before we can definitively conclude this.
>
> > It is not clear why the representations should be mapped to CLIP image features first, as opposed to directly mapping them to a sparse linear combination of words in CLIP (i.e. via sparse autoencoder). A comparison to this approach can be very useful.
>
> The primary reason for this is ease of training, as in this manner, our method can be used on any image dataset without accompanying text descriptions. It also uses a simple cosine distance loss rather than a more complicated loss like contrastive loss. Directly mapping the $c_i$ to the text encoder space also has issues due to the well-known "modality gap" between CLIP image and text representations.
>
> In some experiments that didn't make it into the paper, we trained the maps $f_i$ using a contrastive loss on the MS-COCO dataset. The loss decreased slowly, and the resulting $f_i$ were unsatisfactory in terms of match rate and cosine similarity. One potential reason for this is that the supervision offered by the contrastive loss over text descriptions is much weaker than the cosine similarity with CLIP image features.

---

> > ### Comment · Reviewer_oPeN · 2024-08-08
> >
> > Thank you for the response. I decided to keep my score as is.

---

### Official Review · Reviewer_5CRC · 2024-07-12

**Soundness:** 3
**Presentation:** 3
**Contribution:** 3
**Rating:** 7
**Confidence:** 3

**Summary:**

The paper proposes a novel method for evaluating the contribution of individual components in arbitrary vision transformers and mapping these contributions to the CLIP space for text interpretation. To avoid rigid matching of each component, the paper introduces a continuous scoring function for component-feature pairs. The effectiveness of the proposed method is demonstrated through token-wise contribution heatmaps, retrieval tasks, and the spurious correlations of the Waterbirds dataset.

**Strengths:**

The paper proposes a novel method for interpreting arbitrary vision transformers. It successfully interprets the contribution of each component through text, which could help in understanding the decision-making mechanisms of ViTs.

The paper presents a comprehensive set of experiments, including text-based image retrieval, image-based image retrieval, token contribution heatmaps, and zero-shot spurious correlation analysis. Additionally, ablation studies are conducted to demonstrate the utility of COMPALIGN and the rationale for focusing on the last few layers of the ViT. These experiments robustly validate the effectiveness of the proposed method.

**Weaknesses:**

The paper does not include comparisons with any baseline methods. Although previous methods may not extend to arbitrary Vision Transformer models, a comparison with specific models would be beneficial.

The paper lacks an analysis of failure cases, leaving it unclear how well the proposed method can be applied in real-world scenarios.

**Questions:**

Could you explain why ImageNet pretrained models encode useful features redundantly?

How did you accumulate the contributions from the lower layers into a single contribution $c_{init}$?

Why does component ordering tend to have a higher correlation than feature ordering, particularly for SWIN and MaxVit, as shown in Table 2?

**Limitations:**

The authors have adequately addressed the limitations of the method. It would be beneficial to see an analysis of how other components and blocks contribute to the model.

---

> ### Author Rebuttal · Authors · 2024-08-03
>
> Thank you for your positive review. We are encouraged that you found our work novel with comprehensive experiments. We reply to your questions below:
>
> > The paper does not include comparisons with any baseline methods. Although previous methods may not extend to arbitrary Vision Transformer models, a comparison with specific models would be beneficial.
>
>  Thank you for your suggestion. We have now added results of our experiments on image segmentation with competitive zeroshot segmentation baselines like GradCam and Chefer et al's method in the global rebuttal. Chefer et al's methods are not equipped to work with SWIN and MaxVit, in which case the cells have been left blank.
>
> > The paper lacks an analysis of failure cases, leaving it unclear how well the proposed method can be applied in real-world scenarios.
>
> Some limitations are discussed in the Limitations section in the appendix. When considering the failure cases for real world applications such as spurious correlation mitigation, the most significant is the case when both the spurious features and core features happen to be represented in the same component(s). In this case, it might be necessary to decompose the component contribution a second time into higher order components. However, as we mention in the Limitation section, we do not investigate this possibility currently.
>
> > Could you explain why ImageNet pretrained models encode useful features redundantly?
>
> We tentatively offer the following hypothesis. ImageNet pretrained models (and its components) presumably learn many features that are most useful for datasets in-distribution with ImageNet. Since we conduct our experiments on the ImageNet validation split, it is likely that these models have many more features useful for ImageNet classification compared to other models such as CLIP or DINOv2. It seems that for non ImageNet pretrained models, much of these useful features are concentrated in the last layers. Thus, when performing layer ablation, it so happens that the accuracy drops more quickly when compared to ImageNet pretrained models.
>
> > How did you accumulate the contributions from the lower layers into a single contribution ?
>
> Referring to the figure and the explanation in the global rebuttal, we see the operation of RepDecompose on a single attention mlp block of a vanilla transformer. Here, $z_5$ contains the contributions from all the previous layers. RepDecompose appropriately applies the suitable transforms (in this case, only the LayerNorm) and obtains the contribution of $z_5$ to the final representation $z$. In this case, the contributions from the lower layers are automatically accumulated into $z_5$
>
> > Why does component ordering tend to have a higher correlation than feature ordering, particularly for SWIN and MaxVit, as shown in Table 2?
>
> There are some components whose direct contribution is not useful for encoding any feature at all, although they may have useful indirect effects. When ordering the components according to their ability to retrieve a specific feature, these components are predictably at the bottom. However, for any component, there is no feature which is predictably the most (or least) dominant. Thus, the component ordering problem is inherently a somewhat easier problem, and our proposed scoring function is better at ordering components as compared to features.

---

> > ### Comment · Reviewer_5CRC · 2024-08-10
> >
> > Thanks to the authors for their response. Most of my concerns have been addressed, and I decided to keep my original score.

---

### Author Rebuttal · Authors · 2024-08-02

We thank the reviewers for their extensive and insightful comments. We are encouraged that the reviewers found our work novel, clear, and impactful. We address some common concerns in this global rebuttal:

## How does RepDecompose work?

To illustrate the workings of our algorithm, we describe the steps that the RepDecompose algorithm takes on a simple attention-mlp block of a vanilla ViT transformer. We include a figure of a simplified computational graph in the pdf, please refer to it for the variable names in the following steps.

1. First, we mark (with green borders in the figure in the pdf) the computational nodes in which the contributions of the components get reduced. For the tokenwise contributions, this is the 'matmul' operation, while for the attention head contributions, it is the 'concat' operation. We also detach the graph at the input of each component to stop the algorithm from gathering only the direct contributions and not any higher-order contribution terms arising from the interaction between multiple components.

2. Let the RepDecompose function be denoted by $d(.)$ which takes in a representation and returns an array of contributions. Here, $n$, wherever it appears, is the number of contributions in the decomposition of the input. The $\text{map}(f, d(z))$ operation applies $f$ to every contribution vector in $d(z)$. At each step, it is ensured that the sum of all contribution vectors/tensors in the RHS is equal to the vector/tensor that is being decomposed in the LHS via the distributive property for linear transformations. Then:

   a. $d(z) =\text{map}( \lambda x. \frac{1}{\sigma}(x - \frac{\mu}{n}), d(z_1)) $ (LayerNorm linearized as in Gandelsman et al [1],  $n$ here is the  number of contributions in $d(z_1)$)

   b. $d(z_1) = (d(z_2), d(z_3))$

   c. $d(z_2) = \text{map}(\lambda x. xW_1 + \frac{b_1}{n}, d(z_4))$ ($n$ here is the  number of contributions in $d(z_4)$ )

   d. $d(z_4) = [z_4]$ (stops when it hits a non-linear node)

   e. $d(z_3) = (d(z_5), d(z_6))$

   f. $d(z_6) = \text{map}(\lambda x. xW_o + \frac{b_o}{n}, d(z_7))$ ($n$ here is the  number of contributions in $d(z_7)$ )

   g. $d(z_7) = [[\text{zeropad}(v) \text{ for } v \in u] \text{ for } u \in d(z_8)]$ (Concatenation of a tensor along a dimension can be expressed as a sum of zero-padded tensors)

   h. $d(z_8) = [ [uv \text{ for } (u, v) \in \text{zip}(U.\text{cols}, V.\text{rows}) ]  \text{ for } U \in  d(z_9) \text{ for }  V \in d(z_{10}) ]$ (via the distributive property for matrix multiplication)

   i. $d(z_9) = [z_9]$  (stops when it hits a non-linear node)

   j. $d(z_{10}) = \text{map}(\lambda x. xW_v + \frac{b_v}{n}, d(z_{11}))$ ($n$ here is the  number of contributions in $d(z_{11})$ )

   k. $d(z_{11}) = \text{map}( \lambda x. \frac{1}{\sigma}(x - \frac{\mu}{n}), d(z_{12})) $  ($n$ here is the  number of contributions in $d(z_{12})$ )

   l. $d(z_{12}) = [z_{12}] = [z_5]$ (Stopped since the comp graph is detached, if not the algorithm would return higher-order terms.)

Thus, the final decomposition contains contributions from the MLP, contributions from each attention head from each token via the OV circuit, and $z_5$, all appropriately transformed by the residual transformations like LayerNorm. Note that this is the same process by which the contributions were derived in Gandelsman et al. [1]

## What about real-world applications (like zero-shot segmentation)?

We also present the results of zero-shot segmentation on the Pixel-ImageNet dataset [4], which contains segmentation masks for ImageNet. We compare our method with the two other best performing zeroshot segmentation methods in Gandelsman et al. [1], that is Chefer at al [2] and Gradcam [3]. The implementation of Gradcam is from the `pytorch-grad-cam` library [5]. Our method outperforms the other two methods on all metrics. Note that since both SWIN and MaxVit operate on 32 x 32 patches, their segmentation metrics are worse than the DeiT and DINO models which operate on 16 x 16 patches.

### References:

1. Y. Gandelsman, A. A. Efros, and J. Steinhardt. Interpreting CLIP’s image representation via text- based decomposition. In The Twelfth International Conference on Learning Representations, 2024.

2. Hila Chefer, Shir Gur, and Lior Wolf. Transformer interpretability beyond attention visualization. In Proceedings of the IEEE/CVF Conference on Computer Vision and Pattern Recognition (CVPR)

3. R. R. Selvaraju, M. Cogswell, A. Das, R. Vedantam, D. Parikh and D. Batra, "Grad-CAM: Visual Explanations from Deep Networks via Gradient-Based Localization," 2017 IEEE International Conference on Computer Vision (ICCV)

4. S. Zhang, J. H. Liew, Y. Wei, S. Wei and Y. Zhao, "Interactive Object Segmentation With Inside-Outside Guidance," 2020 IEEE/CVF Conference on Computer Vision and Pattern Recognition (CVPR) (https://github.com/shiyinzhang/Pixel-ImageNet)

5. Jacob Gildenblat and contributors, "PyTorch library for CAM methods" (https://github.com/jacobgil/pytorch-grad-cam)

---

### Author Response · Authors · 2024-08-10
**Gentle Reminder**

We thank all reviewers for taking out their time to give valuable comments and feedback on this work. We are grateful that overall, this paper has been well-received by the reviewers. As the discussion period is coming to an end, we request all reviewers to acknowledge our rebuttal and give us an opportunity to address any lingering concerns, if they haven't done so already. Thank you again for your participation.

---

### Decision · Program_Chairs · 2024-09-25

**Decision:**

Accept (poster)

**Comment:**

This work introduces a general framework to identify and interpret the roles of various components in vision transformers beyond CLIP by decomposing final representations and mapping them to CLIP space for textual interpretation. The framework also includes a novel scoring function to rank component importance, offering insights that enhance tasks like image retrieval, visualizing token importance, and mitigating spurious correlations.

Summary Of Reasons To Publish:

1) Novel method for interpreting arbitrary vision transformers.

2) Extensive and solid experiments validating the effectiveness of the proposed method.

Summary Of Suggested Revisions:

All the major concerns have been addressed by the authors in their rebuttal. I recommend incorporating some of the key clarifications made in the rebuttal into the final version.